# Matrix-based imaging through dynamic scattering

Elad Sunray[1,2], Gil Weinberg [1,2], Benzy Laufer[1,2] & Ori Katz [1,2] ✉

Noninvasive optical imaging through complex scattering media presents a major challenge across multiple fields. State-of-the-art techniques, such as reflection matrix decomposition and neural networks, rely on multiple measurements with varying illumination within the sample decorrelation time, making their application challenging in rapidly varying dynamic media. Here, we show that due to commutativity property of the convolution operation, dynamic scattering in isoplanatic imaging is mathematically analogous to varying illumination in static media. This insight allows leveraging matrix-based approaches developed for static scattering to rapidly varying dynamic media. Specifically, we show that the covariance matrix of a set of scattered light camera frames captured through a dynamic scattering sample has the same mathematical form as the reflection matrix of a static medium, with the target object playing the scattering medium's role. We demonstrate this concept by high-resolution diffraction-limited imaging through dynamic scattering across multiple modalities, from incoherent fluorescence microscopy to coherence-gated holographic reflection imaging.

Optical imaging through scattering media poses a fundamental challenge and opportunity for optical imaging[1,2], and is a field with intense active research and significant recent advancements. Imaging objects through rapidly changing environments, such as atmospheric turbulence[3–5], biological tissues[6,7], or fog[8], holds a unique importance in fields ranging from medical imaging, through remote sensing, to astronomical observations.

Among the recently developed scattering-compensation techniques, reflection-matrix-based methods have emerged as essential tools for noninvasive computational imaging through such media. These approaches rely on dynamic (controlled[9–12] or uncontrolled[13–15]) illumination of a static scene, allowing the measurement of the sample reflection matrix, and subsequent application of scattering compensation and image reconstruction algorithms. As a result of the relatively large number of required consecutive measurements, these techniques are inadequate to tackle rapidly varying scattering, such as those encountered in flowing blood, atmospheric turbulence, or fog.

Recently, approaches based on neural networks or online learning have been proposed to tackle dynamic aberrations or scattering computationally. Neural-network-based approaches include supervised methods[16,17], which heavily rely on training data and usually lack interpretability, and unsupervised methods using neural representations[18,19] that assume slow, correlated temporal variations in the scattering medium, making it challenging to adapt to rapidly uncorrelated media. Online learning of the transmission matrix of dynamic media[20] has been demonstrated, but it too is limited to slowly temporally varying scattering. Impressive efforts to computationally undo dynamic atmospheric turbulence by deep learning have been reported in recent years[3–5,21,22]. However, these methods are specialized in low-order atmospheric aberrations and are not aimed at correcting complex scattering, such as that encountered in biological tissues or highly scattering layers.

Alternatively, speckle-correlation imaging techniques, inspired by stellar speckle interferometry[23], which exploit the statistical properties of speckle patterns to recover image information, have been applied to

[1]Institute of Applied Physics, The Hebrew University of Jerusalem, Jerusalem, Israel. [2]These authors contributed equally: Elad Sunray, Gil Weinberg, Benzy Laufer, Ori Katz. ✉e-mail: orik@mail.huji.ac.il

varying scattering conditions[23–27] and, under certain conditions, can function as single-shot imaging techniques[28–34]. However, despite this advantage, these techniques are hindered by their reliance on iterative phase retrieval[35], which can require a very large number of iterations to converge, as well as specific support priors and potentially a large number of initial guesses. While deterministic bispectrum reconstruction can address the convergence challenge of phase retrieval, it still requires averaging a large number of speckle grains, limiting the reconstruction to relatively simple objects. These limitations underscore the need for an imaging technique that is inherently adapted to image complex scenes through rapidly varying media.

Here, we present an approach that allows us to directly apply state-of-the-art reflection-matrix techniques to dynamic scattering compensation in both coherent and incoherent imaging modalities. Our method overcomes the limitation of a matrix-based approach to static scenes by exploiting the mathematical equivalence of dynamic illumination of a static scene to dynamic scattering under static illumination, leveraging the commutativity property of the convolution model of isoplanatic imaging. Thus, our approach provides a natural and fully interpretable extension of matrix-based imaging techniques to the case of rapidly dynamic scatterers. It enables the reconstruction of complex, megapixel-scale images through rapidly time-varying scattering. Importantly, unlike state-of-the-art neural-networks-based techniques, our approach does not require any assumptions on the temporal variations or other regularization, making it suitable for rapid dynamic scattering.

As our approach is based on a very general principle, it allows versatility in addressing the challenges posed by dynamic scattering media across a wide range of imaging scenarios and modalities. We experimentally demonstrate the approach's efficacy for both incoherent and coherent imaging modalities, including fluorescence microscopy, widefield transmission imaging, and coherent holographic time-gated imaging.

## Results

### Principle

Here, we establish the mathematical foundation for our approach, deriving the mathematical analogy between a dynamic scattering medium and dynamic illumination (Fig. 1).

In isoplanatic (coherent or incoherent) imaging conditions, the image plane distribution is given by a convolution of the object's optical field (in the coherent case) or intensity (in the incoherent case), denoted as $O$, with the effective (field or intensity) point spread function (PSF) $P$:

$$I(\mathbf{r}) = P(\mathbf{r}) * O(\mathbf{r}) \qquad (1)$$

It is important to note that this convolution model is strictly valid only for objects within an isoplanatic patch. All experiments in this work were designed within this constraint. Potential extensions to larger fields of view or thick complex media are discussed in the "Discussion" section.

In the common case where no scattering is present, the PSF is a narrow, sharply peaked function. Consequently, the image on the camera sensor $I$ provides a good direct representation of the object, with a resolution given by the PSF. In the case where isoplanatic scattering or aberrations are present, i.e., in the optical memory-effect range[2], Eq. (1) still holds. However, the scattering PSF is a complex and potentially spatially extended speckle pattern. This results in a low-contrast, blurry, and seemingly information-less image on the camera sensor[2]. The goal of computational scattering compensation is to retrieve the object function, $O$, and potentially the scattering PSF, $P$, without prior knowledge of either $O$ or $P$.

The state-of-the-art techniques for computational scattering compensation rely on measuring the reflection matrix of the sample using a set of controlled[9,10,36] or random[13–15] spatial illumination patterns. The reflection matrix is obtained by multiple recordings,

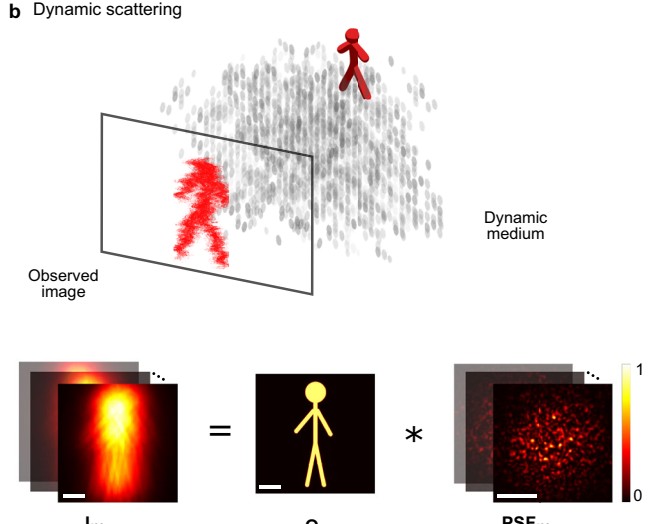

**Fig. 1 | Noninvasive imaging through dynamic scattering, concept.** Our approach leverages matrix-based techniques that were developed for imaging through static scatter using multiple illuminations (**a**), to image through dynamically varying media (**b**). The enabling underlying mathematical principle is the commutativity property of convolution, making the image formation equation in the static-scattering case ((**a**), bottom) mathematically equivalent to the dynamic case ((**b**), bottom), with only the roles of the point spread function (PSF) and the object are interchanged. **a** Conventional matrix-based approaches image through static scattering media by processing a set of captured frames of the scattered light, each obtained by illuminating the object with a different unknown random

illumination[13,14]. In the common case of isoplanatic scattering, each captured image is the convolution of the scattering PSF with the illuminated object ((**a**), bottom). **b** The case of rapidly-varying dynamic scattering poses a challenge for conventional matricial approaches, as multiple captures within the sample decorrelation time are impossible. However, mathematically, for a static object, the captured frames at different times are given by the same convolution equation of the static scattering case (**a**), just with the roles of object and PSF interchanged. Thus, the object and PSF can be reconstructed by applying the conventional matricial algorithms[13,14,57] on the captured frames in the dynamic medium case.

$m = 1... M$, of the scattered complex-valued light field in the coherent case (or scattered light intensity in the incoherent case[14]) under these different illuminations, with each recorded frame in these measurements being expressed as (Fig. 1a):

$$I_m(\mathbf{r}) = P(\mathbf{r}) * O_m(\mathbf{r}) \qquad (2)$$

Where $O_m$ denotes the $m^{th}$ realization of the illuminated object, $O_m(r) = O(r)I_m^{ill}(r)$, and $I_m^{ill}(r)$ is the $m^{th}$ illumination pattern. By arranging these measured images into columns of a matrix $\mathbf{A}$, we can write the measured dataset as $\mathbf{A} = \mathbf{PO}$, where $\mathbf{P}$ is a convolution (Toeplitz) matrix, and $\mathbf{O}$ is a matrix containing the different illuminated object realizations in its columns. Following Lee et al. and Weinberg et al.[13,14], in the case of uncorrelated illuminations, (defined as $\langle \hat{O}_m(r_i)\hat{O}_m(r_j)\rangle_m \propto |O(r_i)|^2 \delta_{i,j}$, where $\hat{O}(r) \overset{\text{def}}{=} O(r) - \langle O_m(r)\rangle_m$), the I-CLASS (Incoherent Closed-Loop Accumulation of Single Scattering) algorithm[14] enables simultaneous retrieval of both $P(r)$ and $|O(r)|^2$ by decomposing the covariance matrix of $\mathbf{A}$, Cov($\mathbf{A}$), to Cov($\mathbf{A}$) = $\mathbf{P}$Cov($\mathbf{O}$)$\mathbf{P}^T$ [13,14].

In dynamic scattering scenarios, where the PSF varies in an uncorrelated manner, but the target object remains relatively unchanged, the state-of-the-art matrix-based approaches fail. However, in such cases, assuming a sufficiently short exposure time, each camera frame (or hologram in the coherent case) (1) can be written as (Fig. 1b):

$$I_m(\mathbf{r}) = P_m(\mathbf{r}) * O(\mathbf{r}) \qquad (3)$$

With $O$ as the static object function and $P_m$ as the $m^{th}$ PSF. Due to the convolution commutativity property, equation (3) can be written as:

$$I_m(\mathbf{r}) = O(\mathbf{r}) * P_m(\mathbf{r}) \qquad (4)$$

Since Equations (4) and (2) have the exact same form, just with the roles of the object and PSF exchanged (Fig. 1), the CTR-CLASS[13] (Compressed Time-Reversal CLASS) or I-CLASS[14] algorithm can be applied on the measurements $I_m(\mathbf{r})$, to allow the efficient extraction of the object $O(\mathbf{r})$. More specifically, in matrix form, arranging the measurements $I_m(\mathbf{r})$ as columns in a matrix $\mathbf{A}$, allows us to write it as $\mathbf{A} = \mathbf{OP}$, where $\mathbf{O}$ is now a Toeplitz matrix with the object function as the convolution kernel, and the columns of $\mathbf{P}$ are the different PSF realizations. For uncorrelated PSF realizations ($\langle \hat{P}_m(r_i)\hat{P}_m(r_j)\rangle_m \propto \delta_{i,j}$, Cov($\mathbf{P}$) is a diagonal matrix, and the CLASS algorithm can be applied on Cov($\mathbf{A}$) (for a discussion of the important case of residual spatial correlations in the covariance matrix see Supplementary Section S2).

Thus, the presented approach is realized by performing the following steps: (1) capture $m = 1... M$ scattered light patterns through a rapidly varying medium; (2) arrange the measurements as columns in a matrix $\mathbf{A}$; (3) apply the I-CLASS[14] algorithm on $\mathbf{A}$ to retrieve the hidden target object (effectively applying the matrix-based adapted CLASS algorithm on the covariance of $\mathbf{A}$).

## Experimental results: incoherent imaging

As a first demonstration, we experimentally demonstrate the effectiveness of our method in conventional transmission imaging through dynamically varying scattering. The optical setup is schematically illustrated in Fig. 2a. A conventional widefield microscope captures images of the various objects through a rotating diffuser illuminated by a spatially incoherent LED illumination at 625 nm central wavelength (see "Methods"). For each imaged object, $M = 150$ short-exposure (0.5–7 ms, see "Methods"), camera frames were captured and processed by the I-CLASS algorithm[14]. The diffuser rotation between captures was such that the (PSFs) of the different captures were uncorrelated (see Supplementary Section S1).

Note that although these initial experiments utilize a transmission geometry with illumination from behind the target, the addition of a scattering medium between the light source and the target in these

experiments would not change the imaging performance as long as a sufficiently high intensity is passed through the scattering medium, as the incoherent imaging configuration only requires homogeneous illumination of the target. Additionally, our subsequent experimental demonstrations in epi-illumination and detection in fluorescence microscopy and coherent holographic imaging demonstrate applicability to noninvasive imaging across diverse optical configurations and imaging modalities.

Figure 2b–q presents the experimental imaging results after applying the I-CLASS reconstruction algorithm. Example raw captured frames of the target objects distorted by the diffuser are given in Fig. 2b, c, f, g, j, k, n, o. As expected, the details of the objects are distorted due to scattering compared to their direct imaging without the diffuser present (Fig. 2e, i, m, q). The reconstructed images obtained by applying the I-CLASS algorithm[14] on the captured frames (after multiplying each frame in Fig. 2b, c, f, g by a fixed scalar value between 1 and 2, see "Methods" and Supplementary Section S2) reveal fine details and features of the objects, are presented in Fig. 2c, g, k, o. The reconstructed object allows estimation of the PSF of each frame by frame-wise deconvolution of each of the raw captured frames with the reconstructed object (Supplementary Section S1). Examples for the full dataset of captured frames, reconstructed objects, and reconstructed PSFs are presented in Supplementary Section S1 and Supplementary Videos S1 and S2.

As a second demonstration, we performed transmission imaging through a naturally-dynamic scatterer. To this end, we replaced the rotating diffuser with a 1 mm-thick cuvette containing 45 μm-diameter Polystyrene beads in solution, creating a dynamically varying scattering as the beads flow freely in the suspension. Additionally, a static diffuser with a 0.5° scattering angle was placed adjacent to the cuvette to ensure no ballistic component was present in the captured scattered light images. The experimental setup and results for these experiments are presented in Fig. 3. Using this configuration, we imaged and reconstructed two resolution test targets. The captured camera frames (Fig. 3b, f, c, g) appear highly blurred, exhibiting no clear features. However, the I-CLASS corrected images (Fig. 3d, h) successfully reconstruct the fine features of the target objects, demonstrating the effectiveness of the approach in correcting such natural dynamic scattering where no ballistic component is present. For reference, direct images of the same resolution targets captured without the scattering medium, using the same widefield transmission microscope, are provided in Fig. 3e, i.

As an additional proof of principle we tested our approach in a fluorescence microscopy experiment performed in epi-detection geometry through dynamic scattering. The setup for this experiment is depicted in Fig. 4a. It is a conventional widefield fluorescence microscope with a rotating diffuser placed between the microscope objective and the fluorescent sample (see "Methods"). The illumination source is a narrowband spatially-incoherent source composed of a 200-mW CW laser (06-MLD-488, Cobolt) and a rapidly rotating diffuser (see "Methods"). An sCMOS camera (Andor Neo 5.5) captures $M = 150$ images of the scattered fluorescence light through a dichroic mirror and appropriate emission filters.

Figure 4 presents the result of the fluorescence microscopy experiments. Figure 4b, e show sample captured frames of two targets composed of fluorescent beads (Fluoresbrite YG microspheres 10 μm), as conventionally imaged through the optical diffuser. The significant distortion due to scattering can be observed in both the raw frames and the zoomed-in areas marked by red-dashed lines. The I-CLASS reconstructed images (Fig. 4c, f), after multiplying each frame by a fixed scalar value (see "Methods" and Supplementary Section S2), successfully recover fine details and features of the objects. For comparison, Fig. 4d, g display the direct images of the objects as imaged without the scattering present.

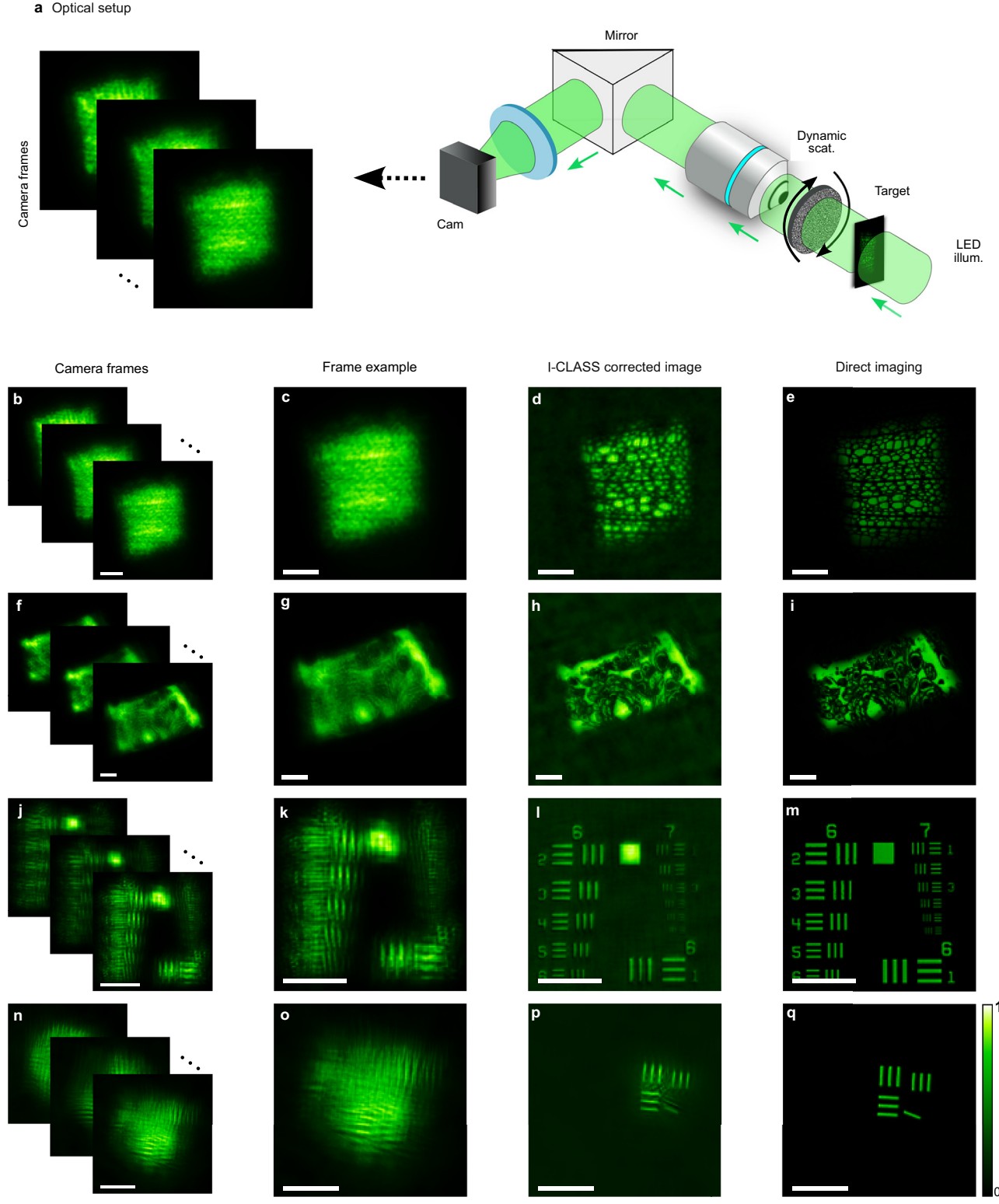

**Fig. 2 | Incoherent imaging proof of principle through a dynamic diffuser.**
**a** Experimental setup: a conventional widefield microscope records $M = 150$ distorted images of incoherently-illuminated targets through a dynamically rotating scattering diffuser. **b, c** Captured raw camera frames. **d** I-CLASS corrected image, revealing the fine details and features of the target. The reconstructed PSF is given in Supplementary Section S1 and Supplementary Movie S1. **e** Reference image of the object as imaged without the diffuser present. **f**–**q** Same as (**b**–**e**) for different target objects and diffusers. Colormaps are scaled between the minimum and maximum values of each reconstruction. Scale bars, 100 μm.

## Experimental results: holographic coherent imaging

As a final demonstration of the generality of the approach and its applicability to various imaging modalities, we applied it to coherent holographic reflective imaging. The results of this study are

presented in Fig. 5. The experimental setup is schematically depicted in Fig. 5a: a reflective target (USAF resolution target) is illuminated by a wide illumination beam through a dynamically rotating diffuser. The illumination is provided by a 632 nm Gaussian beam from a

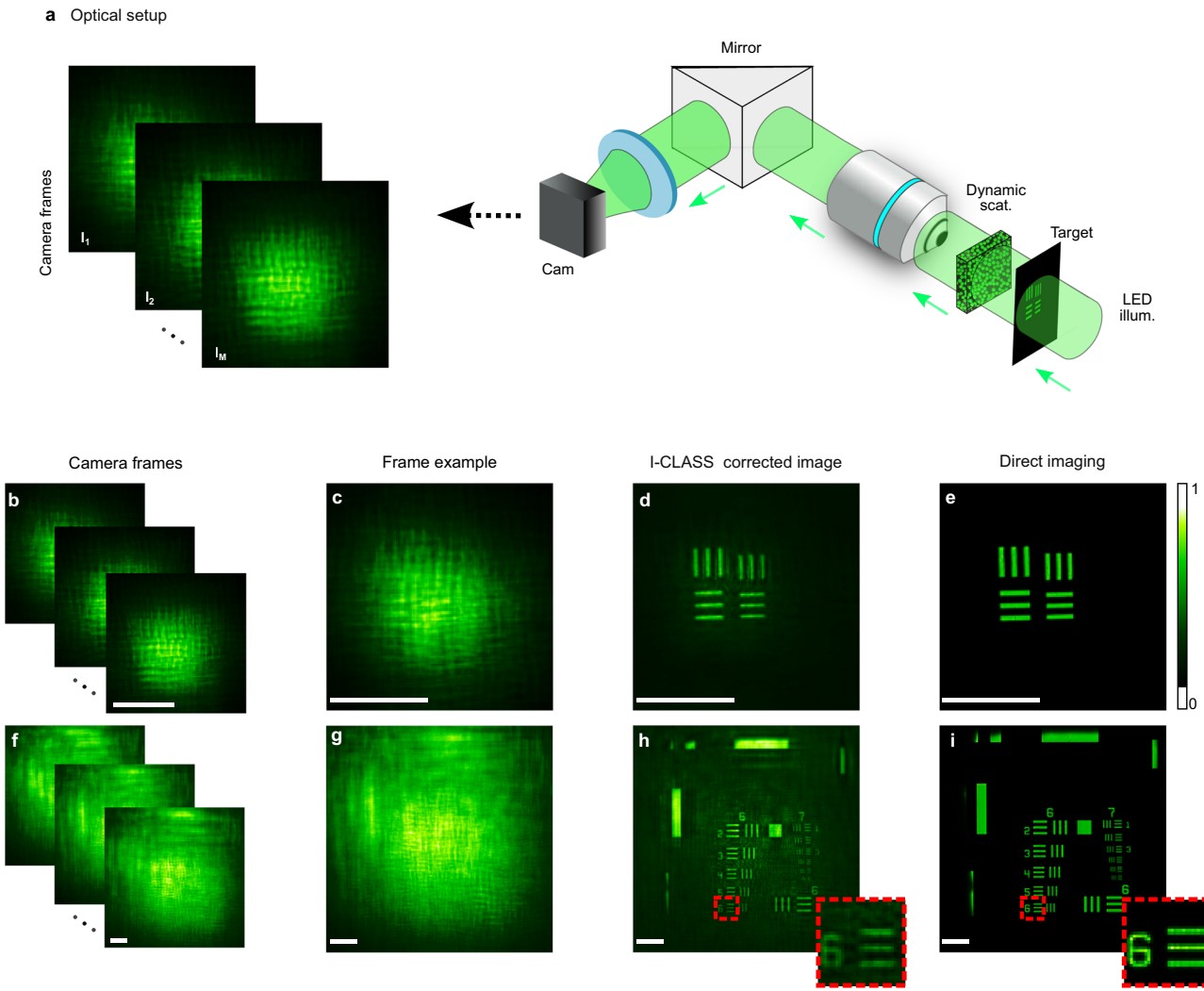

**a** Optical setup

**Fig. 3 | Incoherent imaging through dynamic scattering. a** Experimental setup: a conventional widefield microscope records $M = 150$ distorted images of incoherently-illuminated targets through a 1 mm-thick cuvette filled with a solution of 45 μm polystyrene beads. A static diffuser is added in front of the cuvette to ensure no ballistic component is present. **b**, **c** Experimental camera frames of target imaged through a dynamically rapidly varying medium taken at distinct times. **d** I-CLASS reconstructed images. **e** Images of the object with the cuvette removed. **f**–**i** Same as (**b**–**e**) for a different target object. Insets in **h**, **i** show zoomed-in areas marked by red-dashed lines. Colormaps are scaled between the minimum and maximum values of each reconstruction. Scale bars, 150 μm.

Helium-Neon laser (HNL210L, Thorlabs), which is focused to a tight spot on the diffuser surface to ensure that the object illumination remains relatively constant while the diffuser is varied (see below). An sCMOS camera holographically records $M = 180$ reflected scattered light fields by imaging the diffuser back surface with a 4f imaging system. A reference beam with a proper optical path delay matching the target distance is used for off-axis holographic acquisition[37].

To reconstruct the target, each captured field was digitally propagated to the target plane using Fresnel propagation. As expected, without scattering compensation, the reconstructed fields are highly distorted, and the target object features cannot be observed (Fig. 5b).

If the image-forming equation has the same form as Eq. (4), the CTR-CLASS or I-CLASS algorithms can be directly applied to the holographically captured fields. However, different from the spatially incoherent illumination case (Figs. 2–4), where the illumination at the target plane was homogeneous regardless of the scattering layer dynamics, a significant challenge in the coherent imaging configuration is ensuring a constant illumination of the target despite the dynamic scattering introduced by the rotating

diffuser. Specifically, in the coherent illumination case (Fig. 5), the presence of the dynamic scatterer in the illumination path of a wide beam generates a speckle illumination field at the target plane that may be subject to variations between scatterer realizations. Consequently, the measured field at the camera for the $m^{th}$ realization, $E_m^{cam}(\mathbf{r})$, can be expressed as:

$$E_m^{cam}(\mathbf{r}) = P_m^{coh}(\mathbf{r}) * \left[ O(\mathbf{r}) E_m^{ill}(\mathbf{r}) \right] \quad (5)$$

Where $E_m^{ill}(\mathbf{r})$ is the illumination field at the target plane at the $m^{th}$ realization, $P_m^{coh}(\mathbf{r})$ denotes the complex-valued field amplitude point spread function (APSF) introduced by the scattering medium in the $m^{th}$ realization, and $O(\mathbf{r})$ is the target object spatial reflectivity. From Eq. (5) it is evident that if the illumination pattern remains unchanged across scattering realizations: $E_m^{ill}(\mathbf{r}) = E^{ill}(\mathbf{r})$, then $O(\mathbf{r})E_m^{ill}(\mathbf{r}) = O^{eff}(\mathbf{r})$, may be considered as an effective static object serving as the single object function that is assumed in the model of Eq. (4). To ensure this is the case, we have focused the Gaussian illumination beam on the scattering layer surface such that the beam waist at the scattering layer surface is sufficiently smaller than the correlation length of the

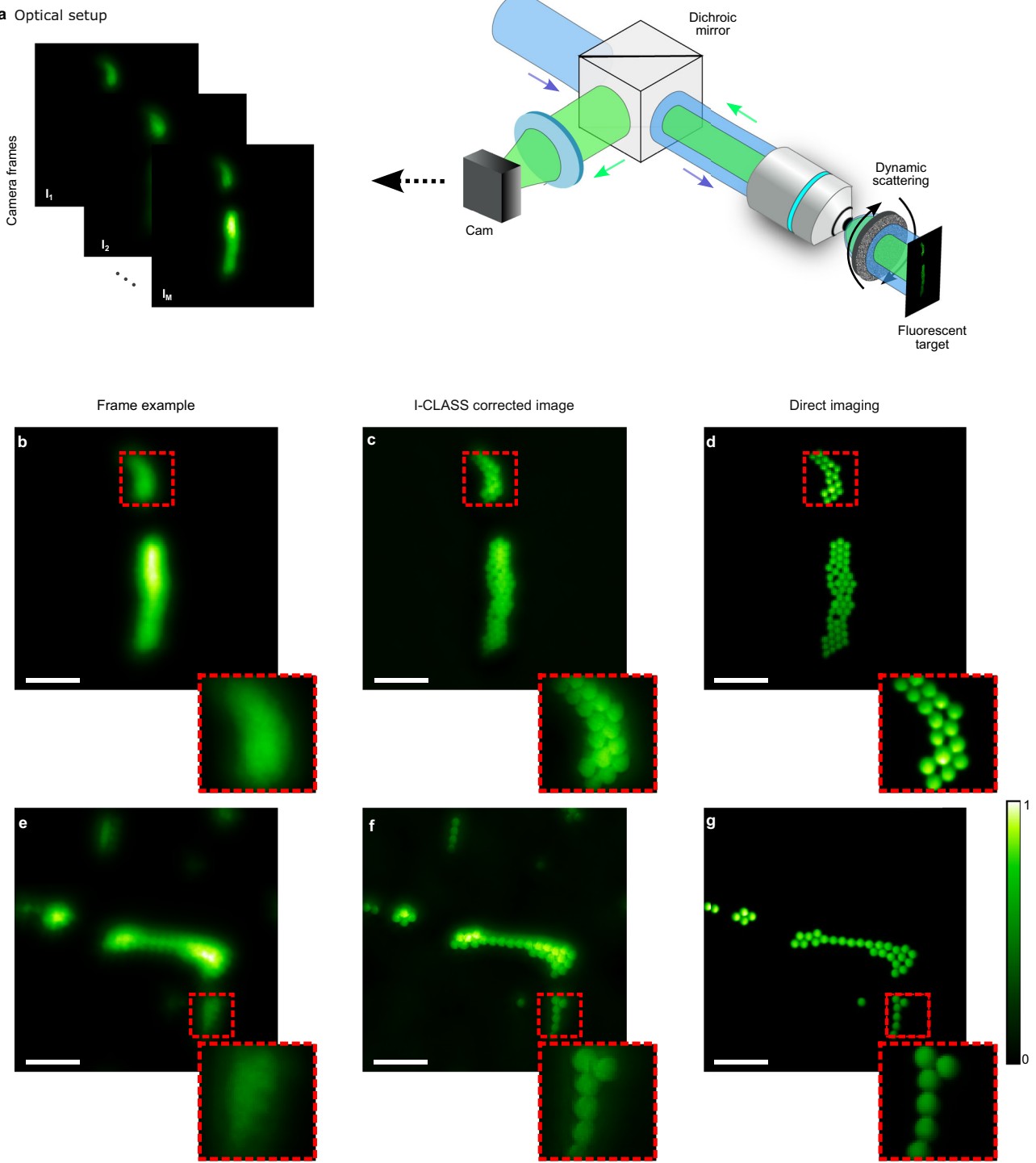

**Fig. 4 | Fluorescence microscopy through dynamic scattering. a** Experimental setup: a conventional widefield fluorescence microscope records $M = 150$ distorted images of fluorescent 10 μm-diameter beads through a dynamically rotating scattering diffuser. **b** Experimental camera frames of fluorescent objects imaged through an optical diffuser, showing distorted images due to scattering. **c** I-CLASS corrected images reveal fine details and features of the objects. **d** Reference images of the objects without scattering layers. **e**–**g** Same as (**b**–**d**) for different target objects. Insets in **b**–**g** show zoomed-in areas marked by red-dashed lines. Color-maps are scaled between the minimum and maximum values of each reconstruction. Scale bars, 100 μm.

scattering layer. This minimizes variations in $E_m^{\text{ill}}(\mathbf{r})$ between realizations, allowing the illumination field to be treated as effectively constant. Under this assumption, the measured fields follow the equation:

$$E_m^{\text{cam}}(\mathbf{r}) \approx P_m^{\text{coh}}(\mathbf{r}) * O^{\text{eff}}(\mathbf{r}) \qquad (6)$$

Since Equations (6) and (4) share the same form, the CTR-CLASS[13] can be applied to the measurements $E_m^{\text{cam}}(\mathbf{r})$, allowing the efficient reconstruction of the object field phase at the scattering layer plane. To reconstruct both the phase and amplitude of the object field, we applied the I-CLASS algorithm[14], which also estimates the object field amplitude from the $M$ captured fields. The I-CLASS

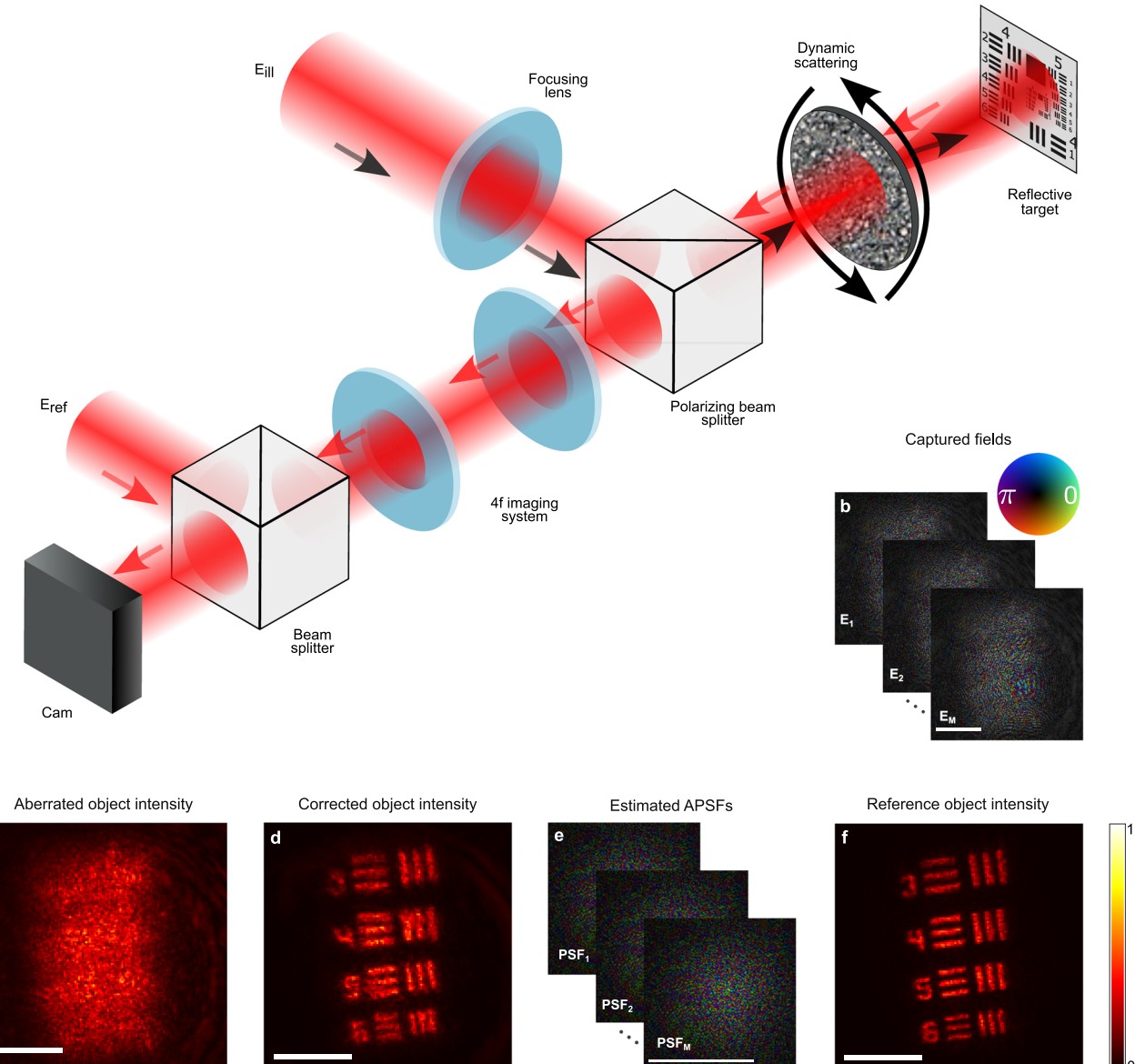

**Fig. 5 | Experimental coherent reflection-imaging through dynamic scattering.**
**a** Experimental setup: a reflective target is illuminated through a dynamically
rotating scattering diffuser. $M = 180$ reflected light fields are holographically
recorded in an off-axis holography configuration using a reference arm. **b** Example
of the recorded distorted fields after computational propagation to the object
plane. **c** One example of the recorded field intensity after computational

propagation to the object plane. **d** Reconstructed object intensity, after applying
the I-CLASS algorithm to compensate for scattering, followed by numerical pro-
pagation to the object plane (see Supplementary Fig. S6). **e** Complex-valued field
amplitude PSFs (APSFs) estimated from each captured field. **f** Reference intensity
image of the object without the diffuser present. Scale bars, 1 mm.

reconstructed object field at the scattering layer plane is numeri-
cally propagated to the target plane by Fresnel propagation to
reconstruct the target, faithfully retrieving the fine features of the
target (Fig. 5d). With the reconstructed target obtained, the APSF for
each frame, $P_m^{\mathrm{coh}}(\mathbf{r})$, can be estimated. This can be achieved by
calculating the phase introduced by the rotating diffuser at each

realization, by taking[15]: $\hat{P}_m^{\mathrm{coh}}(r) = \mathcal{F}\left\{e^{i\hat{\phi}_{\mathrm{diff}}}\right\} = \mathcal{F}\left\{e^{i\arg\left(\frac{E_m^{\mathrm{cam}}(r_{\mathrm{cam}})}{E_{\mathrm{CLASS}}(r_{\mathrm{cam}})}\right)}\right\}$ are

the measured fields at the diffuser plane, and $E_{\mathrm{CLASS}}$ is the CLASS
reconstructed object field at the diffuser plane. As shown in
previous works[15,38], the CLASS reconstructed object field also
contains the spherical phase from the propagation distance

between the object and the scattering layer. Thus, $\hat{\phi}_{\mathrm{diff}}$ provides
an estimation of the diffuser phase. Several estimated APSFs are
shown in Fig. 5e.

## Discussion

We have introduced and experimentally demonstrated a computa-
tional matricial framework for imaging through dynamic scattering
media. The proposed framework addresses an important challenge in
both coherent and incoherent imaging, which conventional matrix-
based approaches have difficulty in tackling due to their reliance on
multiple measurements within the scattering medium decorrela-
tion time.

Importantly, we have shown that under isoplanatic scattering conditions, the covariance matrix of dynamically scattered light fields has the same mathematical form as that of a conventional reflection matrix, with the roles of the medium and target object replaced. Thus, any matrix-based technique capable of decomposing the "reflection-matrix" to an object field and scattering layer can be applied to reconstruct the hidden target. We chose to use the recently introduced I-CLASS algorithm due to the memory-efficient implementation developed in refs. 14,15, which allows high pixel-count processing and also the estimation of the object Fourier amplitude.

Our analysis revealed that energy conservation in phase-only speckle patterns may introduce spatial correlations that can cause background haze in the reconstructions (see Supplementary Fig. S4). Here, we addressed this by simply applying varying intensity scaling across frames during post-processing (see Supplementary Section S2). It would be interesting to study more advanced processing techniques to identify and filter these correlations and/or their effects. In particular, it would be interesting to study matricial processing approaches, such as singular value decomposition filtering (SVD) of the measurement matrix or the covariance matrix. It was recently shown that SVD-based analysis and filtering of the reflection matrix or the distortion matrix[10,39,40], can effectively separate correlated signals, such as those originating from different isoplanatic patches, or that are less affected by noise.

An analysis of the effects of the energy-conservation originated correlations on the reconstruction, and our approach for mitigating them, is detailed in Supplementary Section S2 and Supplementary Fig. S3.

We note that while single-shot speckle-correlation approaches[29] can, in principle, be applied to each of the captured frames since they only use the spatial autocorrelation of a single frame (or an estimation of a single autocorrelation from a set of frames as in stellar speckle interferometry[23]), their performance in terms of reconstruction fidelity for complicated objects and convergence stability are significantly inferior to the proposed covariance-matrix-based approach (see "Numerical Study" in Supplementary Section S3).

We note that for coherent imaging through dynamic scattering, our current approach is limited to scenarios where effectively constant illumination can be maintained at the object plane. While demonstrated here with a thin scattering layer by focusing the illumination to a spot that is smaller than the coherence area of the scattering layer, thick volumetric scattering presents a challenge that requires additional or alternative solutions.

One potential solution for obtaining an effectively homogeneous illumination in the case of coherent illumination through thick dynamic media, while maintaining the capability of coherence gating, without requiring focused illumination, is by digital incoherent compounding the time-gated frames of several illuminations taken within the correlation time of the medium. This utilizes the same principles used in speckle reduction techniques in OCT[41], and was demonstrated for imaging through scattering layers via correlography[31,42,43], however, without leveraging the potential of matrix-based techniques. In such a solution, for each realization of the dynamic scatterer one would: (I) rapidly acquire multiple ($K \gg 1$) coherence-gated holograms under different speckle illuminations (created with an additional diffuser or SLM in the illumination path); (II) Incoherently sum the intensity patterns of the holographically measured fields to incoherently-compounded "macro-frames": $I_m(x,y,z=z_{\mathrm{obj}}) = \sum_{k=1}^{K} |E_{m,k}(x,y,z=z_{\mathrm{obj}})|^2$ that can then be (III) processed using I-CLASS as in our incoherent experiments. This protocol creates effectively uniform illumination through incoherent compounding, while preserving the important coherence-gating capability of coherent light that is crucial for practical reflection-based microscopy and 3D imaging. The $K$ rapid illumination patterns can be random or complementary speckle illuminations that would result in a more homogeneous illumination distribution[44].

Such a hybrid method offers a potential mitigation strategy for thick scattering media at the price of an increase in the number of acquisitions, or equivalently, at the price of acquisition or dynamic scattering speed. Additionally, we highlight the applicability of the tight focusing approach to lensless imaging through highly dynamic flexible multi-core fiber endoscopes, as previously demonstrated only through relatively static fibers[38,45]. In this system, uniform illumination can be obtained by single-mode excitation of a single-fiber core in a relatively straightforward fashion[38,46].

While we have focused our proof-of-principle demonstrations on isoplanatic scattering conditions, extending our approach to thick dynamic scattering media remains an important challenge. Beyond the direct application of mosaicking approaches, which are effective for moderate scattering[13,47], applying a multi-conjugate, "multi-slice" correction[45,48] would be very attractive. However, since in our formulation the dynamic medium plays the role that would normally be occupied by the object in conventional reflection-matrix implementation, and vice versa, the conventional multi-conjugate approach would only address a thick target object rather than a thick scattering medium. Addressing a thick scattering medium thus requires a solution analogous to addressing a thick target object in conventional reflection matrix imaging. Interestingly, the recent approach of Park et al.[49], where thick target objects are considered, may offer a potential path forward. Alternatively, it may be possible to leverage the recent model-based gradient descent optimization approach[45], which can flexibly handle a multi-parameter model, to integrate the case of a thick medium in the dynamic measurements formalism.

Finally, while we have focused our proof-of-principle demonstrations on isoplanatic scattering conditions, the field of view can be extended in anisoplanatic scattering conditions in cases of weakly scattering samples by individually reconstructing and mosaicking different isoplanatic patches[13,15,47,50,51].

In conclusion, this work demonstrates the versatility and universality of matrix approaches to imaging in complex media, extending their applicability from static to dynamic scattering environments.

## Methods

### Experimental setup

Figures 2 and 3 shows the experimental imaging configuration that utilizes an incoherent Thorlabs M625L3 LED light source. This LED emits at a central wavelength of $\lambda = 625$ nm with an output power of 700 mW and a full width at half maximum (FWHM) bandwidth of $\Delta\lambda = 17$ nm. The emitted light covers an extensive area across the target plane located 3 cm away. Image capture is conducted using an Andor Neo 5.5 sCMOS camera, part of a 4f imaging system equipped with a 10× Mitutoyo objective lens (M Plan APO 10X, NA 0.28) and a Thorlabs LA1256-A tube lens (focal length 300 mm). A Thorlabs FBH630-10 band-pass filter, with a central wavelength of 630 nm and a 10 nm FWHM, is employed to filter the incident light.

During the experiments in Fig. 2, the dynamic media was created using a Thorlabs K10CR1 rotation mount with a holographic diffuser 6 mm from the target. For Fig. 4b–g, a Newport 0.5° holographic diffuser was used, and Fig. 2n–q utilized an RPC Photonics EDC-1° diffuser. For Fig. 3 experiments, scattering was introduced via a 1 mm path-length cuvette containing Polystyrene beads (Fluoresbrite YG microspheres, 45 μm) in a solution with a concentration variability $\gg 7\%$. A 0.5° holographic diffuser was attached to the cuvette to eliminate the scatterer's ballistic components. This scatterer was distanced 10 mm for the measurements in Fig. 3b–e and 15 mm in Fig. 3f–i.

The imaging targets varied across experiments. Figure 2b–e, f–i show prepared microscope slides by Maxlapter (Amazon) of willow stem and pine stem respectively. Figures 2j–m and 3f–i show a 3" × 3" Negative 1951 USAF Test Target (R3L3S1N, Thorlabs), and Figs. 2n–q,3b–e featured custom targets on 1.5 mm thick glass slides

coated with Ti (20 nm) and Ag (100 nm) layers, created through E-Beam Lithography.

Figure 4 shows the fluorescence experimental configuration, which consists of a pseudothermal source composed of a 200-mW, 488 nm continuous-wave (CW) laser (06-MLD-488, Cobolt) and a rapidly rotating holographic diffuser (EDC-1˚) at a distance of ≈15 cm from the objective lens. The images were distorted by a discretely rotating holographic diffuser (RD, NEWPORT 0.5˚) using a stepper motor rotation mount (K10CR2, Thorlabs) placed at distances of ≈8 mm from the target object. The images were captured using the same Andor Neo 5.5 sCMOS camera, imaged by a 4f imaging system equipped with a 10× Mitutoyo objective lens (M Plan APO 10X, NA 0.28) and a tube lens (focal length 300 mm, Thorlabs). The light was filtered with a dichroic mirror (DMLP505R, Thorlabs) and an emission filter (MF525-39, Thorlabs). The target consisted of fluorescent beads (Fluoresbrite YG microspheres 10 μm) placed on a cover glass at the objective lens's focal plane.

Figure 5 shows the experimental configuration for the coherent imaging experiments, where holograms were recorded using an off-axis holography setup. A 21-mW polarized CW He-Ne laser (HNL210L, Thorlabs) with a wavelength of 632.8 nm was used for illumination. To split the beam into reference and object paths, a polarizing beam splitter (PBS, PBSW-633, Thorlabs) was employed, with the path difference between the reference and object paths kept within the coherence length of the laser (≈30 cm). After the PBS, the signal beam was rotated using a half-wave plate (WPQ10ME-633, Thorlabs) to match the polarization of the reference beam, allowing them to interfere at the detector. In the object path, the laser beam passed through a Newport 0.5˚ holographic diffuser mounted on a rotating motor (K10CR1, Thorlabs), ensuring uncorrelated scattering patterns due to the diffuser's rotation. A 10× Mitutoyo objective lens (M Plan APO 10X, NA 0.28) focused the illumination beam, ensuring the illumination spot on the diffuser was smaller than the diffuser's correlation length (≈70 μm), maintaining consistent illumination. The negative USAF test target (R1DS1N, Thorlabs) was positioned approximately 7 cm behind the diffuser, with a mirror covered by a diffusive slide placed behind it to simulate a diffusive object. The reflected light traveled back through the diffuser to the camera. A 4f imaging system consisting of two lenses, an $f = 200$ mm (AC508-200-A-ML, Thorlabs) and an $f = 125$ mm (LA1384-A, Thorlabs), was used to image the field at the diffuser plane onto the camera sensor (Thorlabs 8051M-USB), providing a magnification of ×1.6 . A non-polarizing beam splitter (BS03, Thorlabs) was used to recombine the object and reference beams before interfering at the camera plane. Finally, a band-pass filter (MaxLine Laser Line Filter 633) with a center wavelength of 632.8 nm and a bandwidth of 1 nm was positioned in front of the camera to isolate the laser wavelength and reduce noise.

## Experimental parameters

The experimental parameters for the results displayed in Figs. 2–5, including camera exposure times and image pixel counts, are outlined as follows:

The frames in Fig. 2b–e were captured at 1250 × 1250 pixels, each taken at a 0.9 ms exposure time. In Fig. 2f–i, images were captured at a resolution of 1700 × 1700 pixels and then cropped in the Fourier domain to 300 × 300 pixels, with an exposure time of 0.5 ms. In Fig. 2j–m, frames were captured at 700 × 700 pixels with a 1.25 ms exposure time. Figure 2n–q features frames captured at 800 × 800 pixels, cropped in the Fourier domain to 300 × 300 pixels, with an exposure time of 7 ms. For Fig. 3b–e, frames were sized at 800 × 800 pixels, each with a 50 ms exposure time. In Fig. 3f–i, images were captured at a resolution of 1800 × 1800 pixels, first cropped in the Fourier domain to 300 × 300 pixels and then further cropped to 350 × 350 pixels for visualization, with an exposure time of 30 ms per frame. In Fig. 4b–e, frames were captured at 750 × 750 pixels and

cropped in the Fourier domain to 500 × 500 pixels, with an exposure time of 0.275 s per frame. Similarly, in Fig. 4f–j, frames were captured at 650 × 650 pixels, cropped in the Fourier domain to 300 × 300 pixels, with an exposure time of 0.25 s. In Fig. 5, the object frames were initially captured at 850 × 850 pixels and cropped to 350 × 350 pixels for visualization, with an exposure time of 12 ms per frame. Experiments shown in Figs. 2b–m, 3, and 4 utilized $M = 150$ realizations for reconstruction, while results in Figs. 2n–q and 5 used $M = 180$.

We applied intensity modulation by multiplying each camera frame with a fixed scalar factor, linearly varying from 1 to 2 across frames 1–150, to suppress energy conservation-induced correlations in the PSFs for the measurements shown in Figs. 2b–m and 4. This is discussed in more detail in Supplementary Section S2.

The run time of the I-CLASS algorithm on a commercially available GPU (Nvidia RTX4090, 24 GB) was ≈8 ms per iteration for 150 camera frames at a resolution of 300 × 300 pixels and ≈70 ms per iteration for 150 camera frames at a resolution of 850 × 850 pixels. With our standard protocol of 1000 iterations, this yields total processing times of approximately 8 s and 70 s, respectively.

## Fresnel propagation via Fourier-domain transfer function

In Fig. 5, we present the object field located at the physical object plane, $z_{obj}$. However, since, following the principles of conjugate adaptive optics[15,52–54], we measure the fields at the diffuser plane and input these fields to the I-CLASS algorithm, the I-CLASS algorithm reconstructs the complex object field at the same plane. This is since the object field in our dynamic matrix approach is analogous to the scattering medium phase-function that CLASS/I-CLASS algorithms retrieve in the conventional static medium case. We denote this reconstructed object field at the scattering layer plane as $E_o(x, y, z_{scatt})$. To visualize the field at the object plane, $E_o(x, y, z_{obj})$, we back-propagate the reconstructed field from the diffuser plane to the object plane using Fresnel propagation under the paraxial approximation.

This propagation is efficiently implemented in the Fourier domain using the Fresnel transfer function:

$$E_o(x, y, z_{obj}) = \mathcal{F}^{-1}\left\{ \tilde{E}_o(f_x, f_y, z_{scatt}) \cdot H(f_x, f_y; \Delta z) \right\} \qquad (7)$$

Where: $\tilde{E}_o(f_x, f_y, z_{scatt}) = \mathcal{F}\{E_o(x, y, z_{scatt})\}$ is the 2D Fourier transform of the reconstructed field, $H(f_x, f_y; \Delta z)$ is the Fresnel transfer function, $\Delta z = z_{obj} - z_{scatt}$ is the propagation distance, $\mathcal{F}$ and $\mathcal{F}^{-1}$ denote the 2D Fourier and inverse Fourier transforms.

The Fresnel transfer function in terms of spatial frequency is:

$$H(f_x, f_y; \Delta z) = \exp\left[ i\frac{2\pi\Delta z}{\lambda} \right] \cdot \exp\left[ -i\pi\lambda\Delta z(f_x^2 + f_y^2) \right] \qquad (8)$$

Here: $\lambda$ is the illumination wavelength, $(f_x, f_y)$ are the spatial frequency coordinates corresponding to the real-space axes $(x, y)$.

This formulation supports forward and backward propagation by simply changing the sign of $\Delta z$, and is especially suitable for numerical implementation via Fast Fourier transforms.

## Data availability

The experimental data generated in this study have been deposited in the Zenodo repository[55]. All data needed to evaluate the conclusions in the paper are present in the paper and/or the Supplementary Materials.

## Code availability

For applying the I-CLASS algorithm, we used the published memory-efficient implementation described in refs. 14,15 and available at ref. 56. The up-to-date version can be found at https://github.com/Imaging-Lab-HUJI/Fluorescence-Computational-Imaging-Through-Scattering-Layers.

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

## Acknowledgements

This project was supported by the H2020 European Research Council (ERC) grant no. 101002406 (to O.K.). This research was supported by a scholarship sponsored by the Ministry of Science & Technology, Israel (to B.L.).

## Author contributions

O.K. proposed and conceptualized the project with E.S. and G.W.; E.S., G.W., and O.K. designed the incoherent imaging experimental setup. B.L. and O.K. designed the coherent imaging experimental setup. G.W. carried out the incoherent imaging measurements and data analysis with input from E.S.; B.L. carried out the coherent imaging measurements and data analysis. E.S. wrote the reconstruction algorithm code and numerical simulations with input from G.W.; O.K. supervised the project. All authors contributed to the writing of the manuscript.

## Competing interests

The authors declare no competing interests.
