## [Transparent peer review file · Nature Communications]

Matrix-based imaging through dynamic scattering

Corresponding Author: Professor Ori Katz

Version 0:

Reviewer comments:

Reviewer #1

(Remarks to the Author)

Reviewer #2

(Remarks to the Author)

In this manuscript the authors propose to use the I-CLASS algorithm (which was developed in the same group) to deconvolve the unknown object from the (many) unknown point spread function measured through a dynamic scattering medium. This builds on a previous paper from the same group (ref 14, as far as I can see still unpublished).

Does the approach work? Yes, the results in Fig. 2 alone are convincing enough.

Are the results novel? Yes. The algorithm itself is not new, but its application is.

Is the paper written in a way such that somebody knowledgeable in optics but not an ultra-specialist in imaging through scattering media will be able to replicate the results and build up on them? Not really. The main culprit is that the whole CRT-CLASS/I-CLASS are not well-known algorithms, and the only reference given to readers to make themselves familiar with them is not self-contained. This has a very easy fix: add a section in the supplementary information with an explanation of how and why the algorithm works (there is no space restriction there, so there is also no excuse to provide the necessary information).

Another point that is likely to make the life on a non-specialist unnecessarily hard is that eq. 1 is only valid within the isoplanatic patch. For objects larger than the isoplanatic patch nothing of what is presented here will work. This is only briefly addressed at the end of the Discussion section (largely swiping it under the carpet) but if not tackled at the beginning is likely to confuse people.

A few more minor points:

* In the introduction the authors claim that iterative phase retrieval lacks guaranteed convergence, which is technically not true. Convergence might take a VERY long time, but it will eventually happen (see <https://doi.org/10.1364/AO.21.002758>).

* Just below the above statement there are two sentences beginning with "while" which looks like they are the leftover of some copy-paste during editing.

* The results shown in Fig 2 and 3, albeit impressive, require the target to be illuminated from behind, which makes this whole approach invasive. I know the author never explicitly claim non-invasiveness, but they also never make this point clear.

* I might have missed it, but I don't think I have seen any discussion about the time needed for the I-CLASS algorithm to converge and give the claimed results.

* For the results shown in Fig. 5 one needs to know the distance between the object and the scattering medium, as only the field at the scattering layer can be reconstructed, and one needs to know how far to propagate it back.

* I am not sure I understand how focussing the illumination on the scattering medium can reduce the fluctuations of E^{ill}_m . Wouldn't it actually maximise them?

* I am slightly confused by the setup diagram in Fig.5: the role of the polarizing beam splitter near to the scattering medium is clear, as it allows to reject part of the unscattered light, thus maximising the amount of useful signal, but the polarizing beam splitter closer to the camera seems to ensure that the signal and the reference have opposite polarizations, and thus can never interfere. Am I missing something or the diagram is wrong?

* The comparison between I-CLASS and the phase retrieval algorithms in the supplementary information seems weird. The fact that the phase retrieval is shown to never be able to reconstruct any image more complex than a few dots looks to bad to

be true, and is at odd with my personal experience.

In conclusion, this is an interesting paper which needs several changes in how it is presented before it is ready for publication.

Version 1:

Reviewer comments:

Reviewer #1

(Remarks to the Author)

The authors convincingly responded to my comments and made appropriate changes to the manuscript. I favor publication of the manuscript as it is.

Reviewer #2

(Remarks to the Author)

All of my concerns about the manuscript were about clarity, and the authors have done a good job in explaining all the parts that were left unstated or ambiguous in the first submission. I particularly appreciated the new explanation of the CLASS algorithm.

I am now happy to recommend publication.

In this work, the authors extend the matrix approach of optical imaging they have been developing with other groups for the last few years to the context of dynamic scattering media. This topic is of great interest because matrix imaging has shown its great potential for overcoming multiple scattering in the long-standing quest towards deep imaging of complex media such as biological tissues. However, it relies by essence on the hypothesis of a static medium. Their approach relies on the fact that, under the isoplanatic hypothesis, dynamic scattering is mathematically analogous to dynamic illumination in static scattering media. From this analogy, they are then able to use the methods developed in previous papers [13] for scattering compensation under dynamic illumination in which the covariance matrix of the reflected wave-field was investigated rather than just the reflection matrix itself. This covariance matrix has already been recently leveraged to extend the scope of matrix imaging to incoherent (fluorescent) imaging by the authors themselves [14]. This paper is a new important demonstration of the versatility and the universality of matrix approaches to wave imaging in complex media. For incoherent imaging, the demonstration of the superiority of matrix imaging with respect to speckle correlation techniques [28,29] is convincing and might have deserved more publicity than just a short section in the Supplementary Material. For coherent imaging applications, the hypothesis of a constant incident wave-field is a bit contradictory with the problem of dynamic scattering and would deserve more discussion.

As often with this group, this work is technically of excellent quality and its presentation is elegant. The experiments based on rotating diffusers are rather simple and allows the authors to illustrate nicely the concept. However, one can wonder whether the idea proposed by the authors can go beyond imaging through a thin dynamic diffusive layer and to which extent it can be applied to real 3D imaging situation like for biomedical or LIDAR applications. I detail below the main points that, in my opinion, would deserve more discussion before publication.

1/ For instance, in the coherent imaging experiment depicted in Fig.5, the approach requires a constant incident wave-field, which is complicated if one wants to image through a dynamic scattering medium. In the present experiment, the authors circumvent this issue by focusing the incident wave-field in the diffuser plane such that the beam waist is contained in one coherence length of the diffuser. However, in practical situations, the scattering medium is three-dimensional and the condition of a constant incident wave-field becomes impossible to fulfil. In real life, dynamic scattering impact both the incident and reflected wave-fields. The authors do not discuss really this apparent contradiction between the assumption of a constant incident wave-field and dynamic scattering that will break this invariance.

2/ Another issue of three-dimensional scattering is that it breaks the isoplanatic assumption on which the current approach relies. The authors mention in the discussion the idea of mosaicking the field-of-view but it somehow requires a roughly focused incident wave-field and only applies to moderate scattering. Another route is a multi-conjugate adaptive optics scheme in which multiple scattering trajectories are rectified by compensating aberrations from a stack of phase screens conjugated with different planes inside the scattering medium [Kang et al., Nat. Commun., 2023].

Would the current approach proposed by the authors be compatible with such a multi-conjugate strategy? It might be a relevant to discuss this point in the conclusion.

3/ One last point which is not really discussed in the paper is the number of required independent frames M to converge towards a reliable estimation of the object or, in other words, the scaling of the estimator bias with respect to this number M . An analytical formulation of the problem could be performed to derive this number which is important for future implementations of the method. I guess this number will depend on the object complexity/sparsity.

More specific comments:

4/ The supplementary Section S2 that shows the impact of correlations between speckle patterns is very interesting and might have deserved a place in the main manuscript. In this Supplementary Section, these correlations arise from energy conservation (phase-distortions). Maybe a singular value(or equivalently eigenvalue) decomposition of the covariance matrix [10] can help in that case? The first singular vector would be less affected by these negative correlations than the CLASS algorithm. Can the authors comment on that?

5/ The authors show how to circumvent this issue by a multiplication of each frame by a fixed scalar value. This is interesting but the choice of the multiplication factor from one to two lacks of justification. Can the authors provide more details about this choice?

6/ Even if the I-CLASS and CTR-CLASS have already been presented in previous papers, I would suggest to describe them in the Methods section to make the paper self-sufficient.

7/ The same remark holds for the Fresnel operators applied at different stages of the post-processing for the holographic experiment.

8/ In that respect, this is not really clear to me why, in the holographic experiment, the I-CLASS reconstructed object field is obtained in the scattering layer plane? A more detailed description of the method is needed.

Typos:

9/ page 2: "*Thus, enabling a straightforward fully interpretable extension of matrix-based methods to rapidly dynamic scatterers, reconstructing complex megapixel-scale images through rapidly varying scattering. Importantly* »

This is not a sentence, please rephrase.

10/ page 7: "These experiments' experimental setup »

Please rephrase

1 Reviewer #1 Comments (from PDF):

In this work, the authors extend the matrix approach of optical imaging they have been developing with other groups for the last few years to the context of dynamic scattering media. This topic is of great interest because matrix imaging has shown its great potential for overcoming multiple scattering in the long-standing quest towards deep imaging of complex media such as biological tissues. However, it relies by essence on the hypothesis of a static medium. Their approach relies on the fact that, under the isoplanatic hypothesis, dynamic scattering is mathematically analogous to dynamic illumination in static scattering media. From this analogy, they are then able to use the methods developed in previous papers [13] for scattering compensation under dynamic illumination in which the covariance matrix of the reflected wave-field was investigated rather than just the reflection matrix itself. This covariance matrix has already been recently leveraged to extend the scope of matrix imaging to incoherent (fluorescent) imaging by the authors themselves [14]. This paper is a new important demonstration of the versatility and the universality of matrix approaches to wave imaging in complex media. For incoherent imaging, the demonstration of the superiority of matrix imaging with respect to speckle correlation techniques [28,29] is convincing and might have deserved more publicity than just a short section in the Supplementary Material. For coherent imaging applications, the hypothesis of a constant incident wave-field is a bit contradictory with the problem of dynamic scattering and would deserve more discussion.

As often with this group, this work is technically of excellent quality and its presentation is elegant. The experiments based on rotating diffusers are rather simple and allow the authors to illustrate nicely the concept. However, one can wonder whether the idea proposed by the authors can go beyond imaging through a thin dynamic diffusive layer and to which extent it can be applied to real 3D imaging situations like for biomedical or LiDAR applications. I detail below the main points that, in my opinion, would deserve more discussion before publication.

Response

We thank the Reviewer for their general positive evaluation of our work, recognizing its significance in extending matrix-based scattering compensation algorithms to dynamic scattering media, and its potential impact. We agree that the important question on the effects of thick scattering media and applications in coherent imaging deserves more discussion. We address it in our point-by-point replies below.

- 1.

Reviewer Comment

For instance, in the coherent imaging experiment depicted in Fig. 5, the approach requires a constant incident wave-field, which is complicated if one wants to image through a dynamic scattering medium. In the present experiment, the authors circumvent this issue by focusing the incident wave-field in the diffuser plane such that the beam waist is contained in one coherence length of the diffuser. However, in practical situations, the scattering medium is three-dimensional, and the condition of a constant incident wave-field becomes impossible to fulfill. In real life, dynamic scattering impacts both the incident and reflected wave-fields. The authors do not discuss really this apparent contradiction between the assumption of a constant incident wave-field and dynamic scattering that will break this invariance.

Response

We thank the referee for raising this point, which was not sufficiently emphasized in the original manuscript. We recognize that the approach applied for thin scattering layers would indeed not be as effective for volumetric scattering media, where maintaining an effective constant illumination may become challenging.

To address this limitation transparently in our manuscript, we have made the following two revisions to our manuscript: first, we added the following clarification to the Discussion Section:

”We note that for coherent imaging through dynamic scattering, our current approach is limited to scenarios where effectively constant illumination can be maintained at the object plane. While demonstrated here with a thin scattering layer by focusing the illumination to a spot that is smaller than the coherence area of the scattering layer, thick volumetric scattering presents a challenge that requires additional or alternative solutions.”

Secondly, we suggest a potential mitigation technique for obtaining an effectively homogeneous illumination in the case of coherent illumination through thick media, while maintaining the capability of coherence gating, crucial for practical reflection-based microscopy and 3D imaging, and highlight the potential of focused illumination in imaging through dynamically-bent multi-core fibers (MCFs). We have added this discussion to the Discussion section of our manuscript:

”One potential solution for obtaining an effectively homogeneous illumination in the case of coherent illumination through thick dynamic media, while maintaining the capability of coherence gating, without requiring focused illumination, is by digital incoherent compounding the time-gated frames of several illuminations taken within the correlation time

of the medium. This utilizes the same principles used in speckle reduction techniques in OCT [20], and was demonstrated for imaging through scattering layers via correlography [10, 25, 22], however, without leveraging the potential of matrix-based techniques. In such a solution, for each realization of the dynamic scatterer one would: (I) rapidly acquire multiple ($K \gg 1$) coherence-gated holograms under different speckle illuminations (created with an additional diffuser or SLM in the illumination path); (II) Incoherently sum the intensity patterns of the holographically measured fields to incoherently-compounded "macro-frames": $I_m(x, y, z = z_{obj}) = \sum_{k=1}^K |E_{m,k}(x, y, z = z_{obj})|^2$ that can then be (III) processed using I-CLASS as in our incoherent experiments. This protocol creates effectively uniform illumination through incoherent compounding, while preserving the important coherence-gating capability of coherent light that is crucial for practical reflection-based microscopy and 3D-imaging. The K rapid illumination patterns can be random or complementary speckle illuminations that would result in more homogeneous illumination distribution [8].

Such a hybrid method offers a potential mitigation strategy for thick scattering media at the price of an increase in the number of acquisitions, or equivalently, at the price of acquisition or dynamic scattering speed. Additionally, we highlight the applicability of the tight focusing approach to lensless imaging through highly dynamic flexible multi-core fiber (MCF) endoscopes, as previously demonstrated only through relatively static fibers [5, 9]. In this system, uniform illumination can be obtained by single-mode excitation of a single fiber core in a relatively straightforward fashion [5, 27]."

Thirdly, we have performed a numerical study on the effects of imperfect (in)homogeneous illumination on the reconstruction fidelity in the case of coherent imaging with a beam focused at the scattering layer plane. The results of this study are presented in the new Supplementary Sections S5, which reads:

5 Effect of illumination homogeneity and stability on coherent imaging through dynamic scattering

In our coherent imaging demonstration (Fig. 5 of the main text), maintaining a constant homogeneous illumination pattern at the object plane despite the dynamic scattering introduced by the rotating diffuser is important to ensure the 'fixed object' assumption of our model (Eqs. 6-7 of the main text). While obtaining a constant homogeneous illumination through a dynamic scatterer is rather straightforward with spatially-incoherent illumination, it is often challenging when spatially coherent illumination is considered. Nonetheless, it can be achieved for the case of a dynamic *thin* scatterer, such as a diffuser, by focusing the illumination spot size on the diffuser, such that the spot size is sufficiently smaller than the diffuser's coherence (or correlation) area. In this case, the illumination beam effectively experiences propagation through a single coherence area with a nearly constant phase function, and thus does not experience scattering, providing a homogeneous illumination of the target. Fig. S7 demonstrates and numerically studies the limitation of this approach. The top row (Fig. S7a-c) shows the phase pattern of a thin scattering layer with the illumination spot superimposed for three cases: where the illumination spot diameter is 0.1, 0.3, and 0.5 times the diffuser correlation length ($d_{\text{correlation}}$), respectively. The second row (Fig. S7d-f) shows the resulting illumination intensity pattern at the object plane after the light has propagated through the diffuser, simulated using angular spectrum propagation. The third row (Fig. S7g-i) shows the effect of these illumination patterns on the effective object (the product of the object and the illumination), while the fourth row (Fig. S7j-l) shows the reconstruction results.

Figure S7: **Effect of illumination pattern stability on coherent image reconstruction.** Numerical simulations of coherent matrix-based imaging through a thin dynamic scattering layer, studying the effect of illumination stability achieved by focusing the illumination spot size at the scattering layer surface (d_{spot}) to a size that is smaller than the scattering layer correlation length (d_{corr}). Focusing the illumination beam to a size considerably smaller than the coherence area of the scattering layer (a) results in a homogeneous illumination pattern at the target object plane (d) that remains constant while the scattering layer dynamically changes. Each column represents a different ratio of the illumination spot size to the scattering layer correlation length: $\frac{d_{spot}}{d_{corr}} = 0.1$ (left), $\frac{d_{spot}}{d_{corr}} = 0.3$ (middle), and $\frac{d_{spot}}{d_{corr}} = 0.5$ (right). **a-c** The thin scattering layer 'phase screen' pattern (color), with the illumination spot size highlighted. **d-f** Resulting illumination amplitude patterns at the object plane after numerical propagation through the scattering layer and free space. The grayscale colorbar indicates normalized intensity. **g-i** Effective reflected field amplitude at the object plane (i.e., the target object reflectivity profile multiplied by the illumination pattern). **j-l** I-CLASS reconstructed images from 150 dynamic scattering realizations. The small illumination spot size relative to the scattering layer coherence area (left column) maintains a relatively uniform illumination at the object plane, enabling high-quality reconstruction, while larger spot sizes (middle and right columns) result in varying speckle illumination, degrading direct reconstruction quality.

When the illumination spot is much smaller than the scattering layer co-

herence area (Fig. S7a), it effectively experiences a nearly constant phase, providing a relatively uniform illumination at the object plane. This consistency across different realizations of the scattering layer is required to maintain the assumption of our imaging model. Indeed, the I-CLASS reconstruction under these conditions is of rather high quality. Note that there exists a global phase shift of the illumination, but this global phase shift can be mathematically contained in the detection PSF, keeping the effective field reflected from the object fixed.

In contrast, when the illumination spot size is comparable to or larger than the diffuser correlation length (Fig. S7c), the beam simultaneously samples multiple uncorrelated regions of the diffuser. This produces complex speckle patterns at the object plane that vary significantly between diffuser positions, creating a different illumination pattern for each diffuser realization. This variation violates our assumption of constant illumination in Eqs. 6-7 of the main text. As a result, the reconstruction quality gradually degrades as the spot size increases relative to the scattering layer coherence area.

In our experimental implementation described in the main text, we carefully focused the beam to ensure a spot size smaller than the diffuser's correlation length, thereby maintaining sufficiently constant spatial illumination patterns at the object plane across different realizations, as required.

In this discussion, we have added the following references to the revised manuscript:

- [20] - Liba et al., Nature Communications, 2017
- [10] - Idell et al., Optics Letters, 1987
- [22] - Metzler et al., Optica, 2020
- [8] - Gateau et al., Physical Review Letters, 2017
- [9] - Haim et al., Nature Photonics, 2025
- [27] - Weinberg et al., Optics Express, 2024

Reviewer Comment

Another issue of three-dimensional scattering is that it breaks the isoplanatic assumption on which the current approach relies. The authors mention in the discussion the idea of mosaicking the field-of-view but it somehow requires a roughly focused incident wave-field and only applies to moderate scattering. Another route is a multi-conjugate adaptive optics scheme in which multiple scattering trajectories are rectified by compensating aberrations from a stack of phase screens conjugated with different planes inside the scattering medium [Kang et al., Nat. Commun., 2023]. Would the current approach proposed by the authors be compatible with such a multi-conjugate strategy? It might be relevant to discuss this point in the conclusion.

Response

We thank the reviewer for highlighting this important limitation. We have indeed demonstrated reconstruction only under isoplanatic conditions in our current proof-of-principle. The extension to thick scattering media represents an important challenge and direction for future research.

Indeed, in conventional reflection-matrix-based imaging or computational wavefront shaping, thick scatterers can be addressed by either mosaicking, in the case of moderate scattering, or using multi-conjugate approaches leveraging a multi-layer scattering model. However, while mosaicking can still be similarly applied on our datasets, our approach presents a unique twist: in our formulation, the dynamic medium plays the role that would normally be occupied by the object in conventional reflection-matrix implementation, and vice versa. This roles reversal means that the multi-conjugate, multi-slice approach that was recently effectively applied for computational scattering correction would now address a thick target object rather than a thick medium. Addressing a thick scattering medium in our case is thus analogous to addressing a thick target object in conventional reflection matrix works. Interestingly, in this respect, the modeling and treatment of thick targets proposed in the recent work by Park et al. [24] may offer a pathway forward. In this work, the authors present a technique that allows for handling thick target objects rather than treating them as purely planar.

Alternative solutions may also be derived from the model-based gradient descent optimization approach recently reported by Haim et al. [9]. This method's flexibility in optimizing a high number of parameters in a model may be utilized to integrate the case of a thick medium (and thick object).

To address this point in the revised manuscript, we have added the following paragraph to the discussion section:

"While we have focused our proof-of-principle demonstrations on isopla-

natic scattering conditions, extending our approach to thick dynamic scattering media remains an important challenge. Beyond the direct application of mosaicking approaches, which are effective for moderate scattering [19, 23], applying a multi-conjugate, "multi-slice" correction [13, 9] would be very attractive. However, since in our formulation the dynamic medium plays the role that would normally be occupied by the object in conventional reflection-matrix implementation, and vice versa, the conventional multi-conjugate approach would only address a thick target object rather than a thick scattering medium. Addressing a thick scattering medium thus requires a solution analogous to addressing a thick target object in conventional reflection matrix imaging. Interestingly, the recent approach of Park et al. [24], where thick target objects are considered, may offer a potential path forward. Alternatively, it may be possible to leverage the recent model-based gradient descent optimization approach [9], which can flexibly handle a multi-parameter model, to integrate the case of a thick medium in the dynamic measurements formalism."

In this discussion, we have added the following references to the revised manuscript:

- [13] - Kang et al., Nature Communications, 2023
- [9] - Haim et al., Nature Photonics, 2025
- [24] - Oh et al., Nature Communications, 2025

3.

Reviewer Comment

One last point which is not really discussed in the paper is the number of required independent frames M to converge towards a reliable estimation of the object, or, in other words, the scaling of the estimator bias with respect to this number M . An analytical formulation of the problem could be performed to derive this number, which is important for future implementations of the method. I guess this number will depend on the object complexity/sparsity.

Response

We appreciate the reviewer's important question about the number of required independent frames for reliable estimation. To address this systematically, we conducted a comprehensive set of numerical simulations investigating the relationship between the number of realizations (M), the measurement signal-to-noise ratio, the object sparsity, and the reconstruction fidelity. We

present the results of this study in a new supplementary section ("Dependence of reconstruction quality on number of realizations, object sparsity, and SNR").

6 Dependence of reconstruction fidelity on number of realizations, object sparsity, and SNR

To study the dependence of the reconstruction quality on the number of realizations, object complexity, and SNR, we carried an in-depth numerical study where we numerically simulate reconstructions using a different number of random realizations (M) for different object complexities (as given by the object sparsity) for both incoherent and coherent imaging scenarios. The results of these analyses are presented as reconstruction fidelity heatmaps in Fig. S8 To ensure statistical robustness, each data point in these heatmaps represents the average outcome from 10 independent numerical experiments.

Figure S8: **Reconstruction fidelity as a function of the number of realizations, signal to noise (SNR), and object sparsity.** Numerical results studying the reconstruction fidelity as a function of the imaging parameters. Simulated objects (a) consist of a varying number (sparsity) of bright points over a dark background. For each studied case, defined by object sparsity, SNR, and number of realizations, 10 distinct numerical experiments with different random distributions and scattering realizations were simulated and reconstructed. For all simulations, a camera pixel count of $N = 350 \times 350$ was used, and an ~ 18 -pixel wide scattering PSF. Cross-correlation scores between reconstructed and widefield reference images are shown as heatmaps for the different studied scenarios. **a** Sample simulated objects with different sparsity levels: 2^{12} (left) and 2^8 (right) bright points. **b** Results for the incoherent imaging case under different SNR conditions: SNR=5 (left), SNR=10 (middle), and SNR= ∞ (right). **c** Coherent imaging results under the same SNR conditions: SNR=5 (left), SNR=10 (middle), and SNR= ∞ (right). The horizontal axis is the number of realizations (M), and the vertical axis is the object sparsity. The color scale indicates the correlation coefficient from 0 to 1.

These results demonstrate that sparser objects can be reconstructed with fewer realizations, while complex objects need substantially more measurements. Higher SNR conditions predictably improve reconstruction quality, and coherent imaging appears to require an increased number of measurements compared to incoherent imaging for an equivalent fidelity.

We note that the theoretical analysis of CTR-CLASS convergence, as established in previous work

by Lee et al. [19], indicates a logarithmic dependence of the required number of measurements on the degrees of freedom (i.e., the number of camera pixels N). However, this theoretical scaling is modified in practice by additional factors, including object complexity (sparsity) and signal-to-noise ratio (SNR). These factors play important roles in determining the practical minimum number of frames needed for successful reconstruction.

More Specific Comments:

4.

Reviewer Comment

The supplementary Section S2 that shows the impact of correlations between speckle patterns is very interesting and might have deserved a place in the main manuscript. In this Supplementary Section, these correlations arise from energy conservation (phase-distortions). Maybe a singular value (or equivalently eigenvalue) decomposition of the covariance matrix [1](citation [10] in manuscript) can help in that case? The first singular vector would be less affected by these negative correlations than the CLASS algorithm. Can the authors comment on that?

Response

We thank the reviewer for this insightful comment and suggestion.

Regarding the SVD approach: The reviewer raises an interesting point about using singular value decomposition (SVD) to address correlations arising from energy conservation, referencing the work by Badon et al.[1] We note that in this approach, SVD is performed on the distortion matrix rather than on the covariance matrix in our case. While the contexts differ, the principle of using SVD to filter out undesired correlations is valuable. It would be interesting to explore whether a similar decomposition approach could help identify and remove the systematic correlations arising from energy conservation in our measurement matrix. An in-depth study of such advanced filtering techniques for the measurement or covariance matrix will be the focus of future work. Following the reviewer’s suggestion, we have added the following discussion to the main manuscript:

”Our analysis revealed that energy conservation in phase-only speckle patterns may introduce spatial correlations that can cause background haze in the reconstructions (see Supplementary Fig. S4). Here, we addressed this by simply applying varying intensity scaling across frames during post-processing (see Supplementary section 2). It would be interesting to study more advanced processing techniques to identify and filter these correla-

tions and/or their effects. In particular, it would be interesting to study matricial processing approaches, such as singular value decomposition filtering (SVD) of the measurement matrix or the covariance matrix. It was recently shown that SVD-based analysis and filtering of the reflection matrix or the distortion matrix [11, 2, 1], can effectively separate correlated signals such as those originating from different isoplanatic patches, or that are less affected by noise. An analysis of the effects of the energy-conservation originated correlations on the reconstruction, and our approach for mitigating them, is detailed in Supplementary Section S2 and Supplementary Figure S3.”

We maintained the detailed analysis in Supplementary Section S2 as suggested, while bringing the key insight to readers’ attention in the main text.

In this discussion, we have added the following references to the revised manuscript:

- [11] - Jo et al., Science Advances, 2022
- [2] - Badon et al., Science Advances, 2016

5.

Reviewer Comment

The authors show how to circumvent this issue by a multiplication of each frame by a fixed scalar value. This is interesting, but the choice of the multiplication factor from one to two lacks justification. Can the authors provide more details about this choice?

Response

We thank the reviewer for raising this point, which was not properly addressed in the original manuscript. To address this, we added to the revised manuscript a systematic study examining how the choice of modulation depth affects reconstruction quality. The results of this study have been added to the Supplementary Materials section 2, which discusses amplitude modulation.

”As discussed above, we addressed the off-diagonal correlations in the covariance matrix that are introduced by energy conservation by applying variable intensity modulation (scaling) across the captured frames. This was done by simply multiplying each captured frame by a scalar factor. We empirically found that modulating by factors between 1 and 2 gave near-optimal results. In Fig. S8 we present the reconstruction fidelity as a function of the choice of modulation depth for the experimental results of

Fig. 2 in the main manuscript. To generate this result, we have performed multiple reconstructions on the same experimental dataset, where before each reconstruction, a different value of varying amplitude modulation (scaling) was applied. For each depth value, we applied a linear scaling to our experimental data where the multiplication factor for the m -th frame (out of M total frames) was:

$$f_m = 1 + \frac{m-1}{M-1} \cdot (\alpha - 1) = 1, \dots, \alpha \quad (1)$$

Where α controls the total range of modulation with corresponding modulation depth: $\frac{\max(f_m) - \min(f_m)}{\max(f_m) + \min(f_m)} = \frac{\alpha - 1}{\alpha + 1}$.

Figure S4: **Incoherent imaging reconstruction fidelity as a function of post-processing intensity modulation of captured frame.** Analysis of the reconstruction fidelity for the experimental results of Fig.2d as different levels of post-processing intensity modulation are applied before running the I-CLASS reconstruction algorithm. The graph presents the Pearson correlation between the reconstruction and the widefield reference (bottom-right inset) as a function of the applied modulation depth (see text). Insets display sample reconstructions, showing the progressive suppression of background haze as the modulation depth increases. At higher modulation depths, the contrast within the object itself is lowered. The red circle indicates the modulation depth chosen for the incoherent reconstructions.

Our results show that without adding a modulation (modulation depth = 0), the reconstructions display a significant background haze. As the mod-

ulation depth increases, this background haze diminishes progressively. At very high modulation depths, while the background continues to reduce, the contrast is somewhat lowered, which may be the result of an imperfect estimation of the MTF in the I-CLASS algorithm [28]. Modulation depths of 0.3-0.5 used in our main experiments provide an effective trade-off between these competing effects. Additional improvements for addressing this point, such as SVD filtering, will be the focus of future work.

6.

Reviewer Comment

Even if the I-CLASS and CTR-CLASS have already been presented in previous papers, I would suggest describing them in the Methods section to make the paper self-sufficient.

Response

We thank the reviewer for this important suggestion about making the paper self-contained, especially for the non-expert reader. We agree that providing a clear explanation of the I-CLASS algorithm is important for readers outside of the field to understand and potentially implement our approach.

To address this concern while balancing manuscript length considerations, we have added a comprehensive mathematical derivation and explanation of the matricial formalism of the problem of imaging through scattering media, and I-CLASS algorithm to the Supplementary Materials in a new section titled "Matrix-based scattering compensation algorithms." This detailed section:

1. Establishes the mathematical foundations of conventional reflection-matrix imaging
2. Shows how CTR-CLASS/I-CLASS extends this to random illumination scenarios
3. Explains how our approach applies these principles to dynamic scattering through the mathematical duality between object and PSF
4. Details the technical implementation of the memory-efficient algorithm

The mathematical nature and length of this explanation make it more appropriate for the Supplementary Materials rather than the Methods section of the main paper. This approach follows common practice in the field where detailed algorithm derivations are included in supplementary sections to maintain focus on the experimental results and physical insights in the main text. We kindly refer the reviewers to read the supplementary section in full in the Supplementary Materials section S7.

Reviewer Comment

The same remark holds for the Fresnel operators applied at different stages of the post-processing for the holographic experiment.

Response

We thank the reviewer for this suggestion. The revised manuscript now includes a detailed description of the Fresnel propagation operator used in the holographic reconstruction. This has been added to the **Methods** section titled "Fresnel propagation via Fourier-domain transfer function".

Fresnel propagation via Fourier-domain transfer function

In Fig. 5, we present the object field located at the physical object plane, z_{obj} . However, since, following the principles of conjugate adaptive optics [21, 17, 26, 16], we measure the fields at the diffuser plane and input these fields to the I-CLASS algorithm, the I-CLASS algorithm reconstructs the complex object field at the same plane. This is since the object field in our dynamic matrix approach is analogous to the scattering medium phase-function that CLASS/I-CLASS algorithms retrieve in the conventional static medium case. We denote this reconstructed object field at the scattering layer plane as $E_o(x, y, z_{\text{scatt}})$. To visualize the field at the object plane, $E_o(x, y, z_{\text{obj}})$, we back-propagate the reconstructed field from the diffuser plane to the object plane using Fresnel propagation under the paraxial approximation.

This propagation is efficiently implemented in the Fourier domain using the Fresnel transfer function:

$$E_o(x, y, z_{\text{obj}}) = \mathcal{F}^{-1} \left\{ \tilde{E}_o(f_x, f_y, z_{\text{scatt}}) \cdot H(f_x, f_y; \Delta z) \right\} \quad (2)$$

Where: - $\tilde{E}_o(f_x, f_y, z_{\text{scatt}}) = \mathcal{F}\{E_o(x, y, z_{\text{scatt}})\}$ is the 2D Fourier transform of the reconstructed field, - $H(f_x, f_y; \Delta z)$ is the Fresnel transfer function, - $\Delta z = z_{\text{obj}} - z_{\text{scatt}}$ is the propagation distance, - \mathcal{F} and \mathcal{F}^{-1} denote the 2D Fourier and inverse Fourier transforms.

The Fresnel transfer function in terms of spatial frequency is:

$$H(f_x, f_y; \Delta z) = \exp \left[i \frac{2\pi \Delta z}{\lambda} \right] \cdot \exp \left[-i\pi \lambda \Delta z (f_x^2 + f_y^2) \right] \quad (3)$$

Here: - λ is the illumination wavelength, - (f_x, f_y) are the spatial frequency coordinates corresponding to the real-space axes (x, y) .

This formulation supports forward and backward propagation by simply changing the sign of Δz , and is especially suitable for numerical implementation via Fast Fourier Transforms (FFT).

8.

Reviewer Comment

In that respect, it is not really clear to me why, in the holographic experiment, the I-CLASS reconstructed object field is obtained in the scattering layer plane? A more detailed description of the method is needed.

Response

We thank the reviewer for bringing this potentially confusing point to our attention.

As mentioned in the new Methods section discussing the Fresnel propagator, since we measure the field at the scattering layer plane and input this field to the I-CLASS algorithm, the I-CLASS algorithm reconstructs the complex object field (equivalent in the conventional case to the scattering medium phase-function) also at the plane of the scattering layer. The importance of measuring the field and applying CLASS/I-CLASS in the thin scattering layer plane follows the principle of conjugation in conjugate adaptive optics. Conjugate correction involves placing the corrective elements (or, in computational approaches, applying corrections) in a plane that is optically conjugated to the primary source of aberrations. By measuring and applying corrections at the plane where phase (and potentially also amplitude) multiplicative distortions occur (the thin scattering layer plane in our case), one achieves a more efficient isoplanatic correction across the field of view. This concept is well established in adaptive optics microscopy, such as in the works by Mertz et al. [21], and others [17, 26].

In the standard case, the CLASS/I-CLASS algorithm finds the phase distortion at this exact plane where the distortions can be described as multiplicative distortions, i.e., the scattering layer transmission-matrix is a diagonal matrix that effectively describes a thin phase (and in I-CLASS also amplitude) mask. The output of the CLASS algorithm in the standard case, where

the scattering medium is static and the target is dynamically illuminated, is the correction phase mask at the scattering layer plane. In our dynamic-scattering case, the scattering medium and the target change roles: the scattering function changes throughout the measurements while the object remains fixed. Running the I-CLASS algorithm on our dataset thus results in an output that is the object phase and amplitude function at the scattering layer plane. This complex-valued field is then back-propagated to any desired distance until a sharp, focused image of the target is obtained (see new Supplementary Fig. S6).

To address this point in the revised manuscript, we have added the following explanation to the **Methods** section, where we discuss the Fresnel reconstruction of the object field at the object plane:

”...since, following the principles of conjugate adaptive optics [21, 17, 26, 16], we measure the fields at the diffuser plane and input these fields to the I-CLASS algorithm, the I-CLASS algorithm reconstructs the complex object field at the same plane. This is since the object field in our dynamic matrix approach is analogous to the scattering medium phase-function that CLASS/I-CLASS algorithms retrieve in the conventional static medium case. ”

In this discussion, we have added the following references to the revised manuscript:

- [21] - Mertz et al., Applied Optics, 2015
- [17] - Kwon et al., Nature Communications, 2023
- [16] - Katz et al., Nature Photonics, 2012

Typos:

9.

Reviewer Comment

Page 2: “Thus, enabling a straightforward fully interpretable extension of matrix-based methods to rapidly dynamic scatterers, reconstructing complex megapixel-scale images through rapidly varying scattering. Importantly. . .” This is not a sentence, please rephrase.

Response

We apologize for this phrasing mixup. We have revised the statement to read:

”Thus, our approach provides a natural and fully interpretable extension of matrix-based imaging techniques to the case of rapidly dynamic scat-

terers. It enables the reconstruction of complex, megapixel-scale images through rapidly time-varying scattering. Importantly, unlike state-of-the-art neural-networks-based techniques, our approach does not require any assumptions on the temporal variations or other regularization, making it suitable for rapid dynamic scattering.”

10.

Reviewer Comment

Page 7: “These experiments’ experimental setup...” Please rephrase.

Response

We have revised the sentence to read:

”The experimental setup and results for these experiments...”

which removes the redundancy and improves clarity.

2 Reviewer #2 Comments (from email):

In this manuscript the authors propose to use the I-CLASS algorithm (which was developed in the same group) to deconvolve the unknown object from the (many) unknown point spread function measured through a dynamic scattering medium. This builds on a previous paper from the same group (ref 14, as far as I can see still unpublished).

Does the approach works? Yes, the results in Fig. 2 alone are convincing enough.

Are the result novel? Yes. The algorithm itself is not new, but its application is.

Is the paper written in a way such that somebody knowledgeable in optics but not an ultra-specialist in imaging through scattering media will be able to replicate the results and build up on them? Not really.

1.

Reviewer Comment

The main culprit is that the whole CRT-CLASS/I-CLASS are not well-known algorithms, and the only reference given to readers to make themselves familiar with them is not self-contained.

This has a very easy fix: add a section in the supplementary information with an explanation of how and why the algorithm works (there is no space restriction there, so there is also no excuse to provide the necessary information).

Response

We thank the reviewer for raising this important point and for the constructive suggestion. Following the referee's proposal, we have added a comprehensive explanation of the I-CLASS algorithm to make the paper self-contained and accessible to readers who may not be specialists in imaging through scattering media, and to the potentially large readership that is not familiar with the recent advancement of matrix-based algorithms. To this end, we have added a new supplementary section ("I-CLASS and matrix-based scattering compensation algorithms") that provides a complete mathematical foundation and derivation of:

1. The reflection-matrix formulation of conventional imaging
2. Reflection-matrix-based scattering compensation algorithm known as CLASS for phase-only correction.
3. The extension of the reflection-matrix acquisition to random illuminations and the application of CTR-CLASS
4. The extension and application of CLASS to dynamic scattering through the exchange of

roles between the object function and the scattering medium function

5. Explanation of the I-CLASS algorithm for phase and amplitude correction.

This new supplementary section reads:

7 Matrix-based scattering compensation algorithms

In this section, we provide the background and mathematical foundation for the I-CLASS algorithm used to reconstruct the target objects in our work. The I-CLASS algorithm builds upon the fundamental principles of reflection-matrix-based imaging techniques [19]. To understand its operation, we first establish the mathematical foundation of the reflection-matrix formalism for imaging through static scattering media, and explain the basics of the original CLASS algorithm, applied to imaging with deterministic controlled illuminations, such as plane waves or scanning beams [12, 29]. We then proceed to show how the matricial formalism and the CLASS algorithm were extended to the case of random illuminations via CTR-CLASS [19], and the extension to phase and amplitude correction by I-CLASS [28]. Finally, we explain how these principles apply to dynamic scattering by the exchange of roles between the scattering medium and the target object.

Reflection-matrix formalism

We begin by considering the simplest scenario of coherent imaging a reflective planar target object through a scattering layer. For simplicity, we assume isoplanatism, i.e., that the imaged field is a convolution of the field at the object plane by a complex-valued field point spread function (PSF). This is indeed the case when imaging objects that are smaller than the isoplanatic patch size [3, 28]. Under these conditions, the measured output field is given by:

$$E_{out}(\vec{r}) = P_{det}(\vec{r}) * E_{obj}(\vec{r}) \quad (4)$$

where $E_{obj}(\vec{r})$ is the reflected field at the object plane, and $P_{det}(\vec{r})$ represent the detection PSF, which is the result of the combined effect of the scattering medium and the imaging system, and $*$ denotes convolution.

The field reflected from the object at the object plane, $E_{obj}(\vec{r})$, is the product of the object reflectivity $O(\vec{r})$ and the illumination at the object plane. Since the considered scenario is of a target object that is hidden behind a thin scattering layer, the illumination at the target object plane is given by a convolution of the 'input' illumination field without the scattering medium present, $E_{in}(\vec{r})$, and the effective 'illumination PSF', $P_{ill}(\vec{r})$, that results from the scattering medium and potentially also the illumination system. Thus, the measured output field behind the scattering layer when an input illumination $E_{in}(\vec{r})$ is used is given by:

$$E_{out}(\vec{r}) = P_{det}(\vec{r}) * [O(\vec{r}) \cdot (P_{ill}(\vec{r}) * E_{in}(\vec{r}))] \quad (5)$$

When sampling these fields and PSFs on a discrete grid with N pixels, and arranging the fields into column vectors with N entries each, the linear relationship given by Supp.Eq. 5 above can be expressed in matricial form as:

$$\vec{E}_{out} = \mathbf{P}_{det} \mathbf{O} \mathbf{P}_{ill} \vec{E}_{in} \equiv \mathbf{R} \vec{E}_{in} \quad (6)$$

Here, \mathbf{P}_{ill} and \mathbf{P}_{det} are $N \times N$ Toeplitz (convolution) matrices representing the illumination and detection PSFs (with the shifted PSFs as their columns), and \mathbf{O} is a diagonal matrix with the object complex-valued reflectivity on its diagonal. The product of the illumination matrix, object matrix, and detection matrix describes the propagation of any input field through the scattering medium to the object and back from the object through the scattering medium to the detection system. This matrix product forms the reflection matrix of the complex medium and the target object, which has a characteristic Toeplitz-Diagonal-Toeplitz (TDT) structure:

$$\mathbf{R} \equiv \mathbf{P}_{det} \mathbf{O} \mathbf{P}_{ill} \quad (7)$$

For a static scattering medium and object scenario, one can measure the reflection matrix, \mathbf{R} , column by column by illuminating the medium with

$m = 1..N$ controlled input fields (basis vectors,) and recording the output scattered light fields for each of these $m = 1..N$ modes, which in matricial form translates to:

$$\vec{E}_{out,m} = \mathbf{R}\vec{E}_{in,m} \quad (8)$$

Once the reflection matrix is measured, the challenge of reconstructing the image of the hidden object from the scattered light measurements is equivalent to decomposing the reflection matrix into its three components. The CLASS algorithm [5] is fundamentally a matrix decomposition method that can take any matrix with this TDT structure and decompose it into its constituent matrices. This allows recovery of the aberration-free object reflectivity function, \mathbf{O} , by separating it from the distortions introduced by the PSFs. Importantly, such a decomposition is made possible due to the fact that while the full reflection matrix contains $N \times N = N^2$ elements (measurements), it is the product of three matrices, where in each matrix there are only N unknown elements. Thus the challenge is to retrieve the $3N$ unknowns (target object and two PSFs) from N^2 measurements.

The CLASS scattering-compensation algorithm

The CLASS algorithm [12, 14] decomposes a reflection matrix given by: $\mathbf{R} = \mathbf{P}_{det}\mathbf{O}\mathbf{P}_{ill}$ that possess a Toeplitz–Diagonal–Toeplitz (TDT) product structure, to the three matrices that compose it. Thus, simultaneously recovering, in an iterative manner, the object and the illumination and detection isoplanatic distortions.

The process begins by Fourier transforming the reflection matrix from real space to the Fourier space coordinates. Since in real space the reflection matrix has a Toeplitz–Diagonal–Toeplitz (TDT) product, after a two-dimensional Fourier transform, the reflection matrix in Fourier space coordinates possess a Diagonal–Toeplitz–Diagonal (DTD) structure:

$$\tilde{\mathbf{R}} = \tilde{\mathbf{P}}_{det} \tilde{\mathbf{O}} \tilde{\mathbf{P}}_{ill} \quad (9)$$

The diagonal structure of the illumination and detection isoplanatic distor-

tions matrices in Fourier coordinates is easily understood as the complex amplitude transfer function (or the OTF in incoherent imaging) of the scattering medium, i.e. the phase of a thin phase-mask scattering layer model. The CLASS algorithm operates under the assumption that both $\tilde{\mathbf{P}}_{\text{ill}}$ and $\tilde{\mathbf{P}}_{\text{det}}$ represent pure phase-only masks, i.e. $\tilde{\mathbf{P}}_{\text{ill}} = \text{diag}(e^{i\phi_1}, e^{i\phi_2}, \dots)$, where ϕ_k denotes the illumination phase distortion at the k -th Fourier component, and similarly for $\tilde{\mathbf{P}}_{\text{det}}$ with the detection-path phase distortions. We note that the assumption of phase-only distortions is relaxed in the recent I-CLASS algorithm [28], explained below.

To illustrate the basic working principle of CLASS, we present a simple 3×3 toy-model example that can be easily extended to the general $N \times N$ case:

1. Setting up the toy matrices. We define our matrices as:

$$\tilde{\mathbf{P}}_{\text{ill}} = \text{diag}(e^{i\phi_1}, e^{i\phi_2}, e^{i\phi_3}), \quad \tilde{\mathbf{O}} = \begin{bmatrix} \tilde{O}_1 & \tilde{O}_3 & \tilde{O}_2 \\ \tilde{O}_2 & \tilde{O}_1 & \tilde{O}_3 \\ \tilde{O}_3 & \tilde{O}_2 & \tilde{O}_1 \end{bmatrix},$$

and $\tilde{\mathbf{P}}_{\text{det}} = \text{diag}(e^{i\psi_1}, e^{i\psi_2}, e^{i\psi_3})$.

2. Alternating correction approach. The key insight that is the basis for the CLASS algorithm is that the multiplication of $\tilde{\mathbf{O}}$ from the right by the diagonal matrix $\tilde{\mathbf{P}}_{\text{ill}}$ results in the multiplication of each k -th column of $\tilde{\mathbf{O}}$ by a single scalar value that is the phase distortion of the k -th component. In a similar fashion, the multiplication of $\tilde{\mathbf{O}}$ from the left by the diagonal matrix $\tilde{\mathbf{P}}_{\text{det}}$ simply multiplies each row of $\tilde{\mathbf{O}}$ by a different scalar value that represents the detection distortions of this k -vector.

CLASS iteratively finds these scalar values (i.e. Fourier space phase distortions) by calculating the correlations between (e.g. neighboring) columns and rows. It is based on alternately correcting the illumination and detection distortions, where in each step it considers illumination-only distortions or detection-only distortions. Correcting one set of distortions, while temporarily ignoring the others.

For instance, to correct $\tilde{\mathbf{P}}_{\text{ill}}$, CLASS conceptually treat $\tilde{\mathbf{R}}$ as if it were simply $\tilde{\mathbf{O}}\tilde{\mathbf{P}}_{\text{ill}}$. In our toy-model example, this product is:

$$\tilde{\mathbf{O}}\tilde{\mathbf{P}}_{\text{ill}} = \begin{bmatrix} \tilde{O}_1 e^{i\phi_1} & \tilde{O}_3 e^{i\phi_2} & \tilde{O}_2 e^{i\phi_3} \\ \tilde{O}_2 e^{i\phi_1} & \tilde{O}_1 e^{i\phi_2} & \tilde{O}_3 e^{i\phi_3} \\ \tilde{O}_3 e^{i\phi_1} & \tilde{O}_2 e^{i\phi_2} & \tilde{O}_1 e^{i\phi_3} \end{bmatrix} \quad (10)$$

3. Column-shifting operation. A critical step in CLASS is to recognize that, due to the Toeplitz structure of $\tilde{\mathbf{O}}$, one can shift the columns of the matrix to align the object's Fourier-components. Specifically, we shift each column so that each row contains the same object component multiplied by a different phase factor:

$$(\tilde{\mathbf{O}}\tilde{\mathbf{P}}_{\text{ill}})_{\text{shifted}} = \begin{bmatrix} \tilde{O}_1 e^{i\phi_1} & \tilde{O}_1 e^{i\phi_2} & \tilde{O}_1 e^{i\phi_3} \\ \tilde{O}_2 e^{i\phi_1} & \tilde{O}_2 e^{i\phi_2} & \tilde{O}_2 e^{i\phi_3} \\ \tilde{O}_3 e^{i\phi_1} & \tilde{O}_3 e^{i\phi_2} & \tilde{O}_3 e^{i\phi_3} \end{bmatrix} \quad (11)$$

This alignment is the key that allows CLASS to extract the phase distortions by cross-correlating the different matrix columns.

4. Extracting phase information. To extract the phase distortions, CLASS performs two operations:

a) Calculate the mean of each row to obtain a vector \vec{T} :

$$\vec{T} = \begin{bmatrix} \tilde{O}_1 \cdot \frac{1}{3}(e^{i\phi_1} + e^{i\phi_2} + e^{i\phi_3}) \\ \tilde{O}_2 \cdot \frac{1}{3}(e^{i\phi_1} + e^{i\phi_2} + e^{i\phi_3}) \\ \tilde{O}_3 \cdot \frac{1}{3}(e^{i\phi_1} + e^{i\phi_2} + e^{i\phi_3}) \end{bmatrix} = \begin{bmatrix} \tilde{O}_1 M \\ \tilde{O}_2 M \\ \tilde{O}_3 M \end{bmatrix} \quad (12)$$

where $M = \frac{1}{3}(e^{i\phi_1} + e^{i\phi_2} + e^{i\phi_3})$ represents a mean phase term.

b) Calculate the scalar product of each of the matrix columns with the vector \vec{T} to find their relative phase shift. This is performed in a matrixial notation by multiplying the transpose of the shifted matrix by the conjugate of \vec{T} :

$$\vec{v} = (\tilde{\mathbf{O}}\tilde{\mathbf{P}}_{\text{ill}})_{\text{shifted}}^T \vec{T}^* = S \cdot |M| \cdot e^{-i\theta_M} \begin{bmatrix} e^{i\phi_1} \\ e^{i\phi_2} \\ e^{i\phi_3} \end{bmatrix} \quad (13)$$

where $S = |\tilde{O}_1|^2 + |\tilde{O}_2|^2 + |\tilde{O}_3|^2 > 0$ is a positive real scalar and θ_M is the

phase of M .

The phase angle of each element in \vec{v} now gives us:

$$\arg(\vec{v}) = \begin{bmatrix} \phi_1 - \theta_M \\ \phi_2 - \theta_M \\ \phi_3 - \theta_M \end{bmatrix} \quad (14)$$

This provides the relative phases of $\tilde{\mathbf{P}}_{ill}$ up to a global phase offset θ_M , which is physically irrelevant as only relative phases matter for image reconstruction.

5. Applying correction for the illumination distortions. In this step, each matrix column is multiplied by the conjugate of the phase distortion, to 'align' all columns. This is performed in matricial form by creating a correction matrix using the phase distortions estimated in the previous step:

$$\tilde{\mathbf{P}}_{ill}^{\text{correction}} = \text{diag}(e^{-i \arg(v_1)}, e^{-i \arg(v_2)}, e^{-i \arg(v_3)}) \quad (15)$$

And applying this correction to the reflection matrix columns:

$$\tilde{\mathbf{R}}_{\text{corrected}} = \tilde{\mathbf{R}} \cdot \tilde{\mathbf{P}}_{ill}^{\text{correction}} \quad (16)$$

6. Detection phase distortions correction. After correcting the illumination distortion, CLASS performs the detection distortion by taking the transpose of the corrected reflection matrix, and repeating the same steps (3-5) to correct the detection distortions, as the rows and columns are now switched. $\tilde{\mathbf{P}}_{det}^T$ now plays the role previously held by $\tilde{\mathbf{P}}_{ill}$, allowing us to apply the same procedure to correct the detection distortion.

7. Iterative refinement. The illumination and detection correction steps are alternated iteratively until the phase estimates converge. Convergence of this process under single-scattering conditions is proven in [12], and usually requires a few tens of iterations.

8. Reconstruction and extraction. After convergence, applying the final phase correction to all matrix columns (and rows) results in an aberration-free reflection matrix, where each column represents an aberration-free

frame. One can then use any single frame as the reconstruction or coherently compound the different frames to reconstruct an improved image, which is equivalent to reconstructing a confocal image from plane wave illumination.

CTR-CLASS and I-CLASS: retrieving a reflection matrix from measurements using unknown input fields

In the case that the input fields for measuring the reflection matrix cannot be controlled or are unknown, one can still measure the resulting scattered light field and retrieve a 'virtual reflection matrix' from the covariance matrix of these fields. This concept for 'compressive time-reversed' (CTR) measurement of the reflection matrix was introduced by Lee et al. [19], and forms the basis for what is termed a CTR-CLASS approach. In CTR-CLASS, instead of acquiring the full matrix by N measurements, one illuminate the object with $M \leq N$ different random and unknown fields, $S_m(\vec{r}) = P_{ill} * E_{in,m}(\vec{r})$. For each illumination, the field equation becomes:

$$E_{out,m}(\vec{r}) = P_{det}(\vec{r}) * [O(\vec{r})S_m(\vec{r})] = P_{det}(\vec{r}) * O_m(\vec{r}) \quad (17)$$

where $O_m(\vec{r}) = O(\vec{r})S_m(\vec{r})$ represents the m-th reflected field from the object. Arranging these M measurements as columns of a 'measurement matrix', \mathbf{A} , we have:

$$\mathbf{A} = \mathbf{P}_{det} \mathbf{O} \mathbf{S} \quad (18)$$

Where \mathbf{S} is a matrix that contains the illumination patterns at the object plane, $S_m(\vec{r})$, as its columns.

Importantly, if the illumination patterns are random and uncorrelated, the covariance matrix of \mathbf{A} takes the form:

$$\mathbf{A} \mathbf{A}^\dagger = \mathbf{P}_{det} \mathbf{O} (\mathbf{S} \mathbf{S}^\dagger) \mathbf{O}^\dagger \mathbf{P}_{det}^\dagger \approx \mathbf{P}_{det} |\mathbf{O}|^2 \mathbf{P}_{det}^\dagger \quad (19)$$

where we have used the fact that $\mathbf{S} \mathbf{S}^\dagger \approx \mathbf{I}$ for uncorrelated illuminations.

The result of Supp.Eq. 19 shows that the covariance matrix of scattered light fields obtained under random unknown illumination has the same TDT structure as the conventional reflection matrix (Supp. Eq. 7) with $|O|^2$ replacing O . Thus, the CLASS algorithm can be applied directly to this covariance matrix to retrieve both the absolute value of the object reflectivity from $|O|^2$, and the detection PSF, P_{det} .

While the CLASS algorithm was developed to compensate for phase-only distortions, i.e. to the case where the illumination or detection PSF are each given by a Fourier transform of a phase-only mask, the recently introduced I-CLASS algorithm [28] extends the CLASS correction (or equivalently, matrix decomposition) to the case where both amplitude and phase distortions exists.

Dynamic scattering correction: exchange of roles between the target object and the scattering PSF

In the dynamic scattering scenario considered in this work, the fundamental imaging equation (Eq. 17) changes. Instead of considering the static scattering medium (PSF) and $m = 1..M$ varying illuminations of the conventional reflection matrix acquisition, which can be described by:

$$E_{out,m}(\vec{r}) = P_{det}(\vec{r}) * O_m(\vec{r}) \quad (20)$$

where $O_m(\vec{r})$ is the field reflected from the object at the m -th illumination, we consider a static object imaged by a time-varying PSF that results from the dynamics of the scattering medium. The imaging equation in this case becomes:

$$E_{out,m}(\vec{r}) = P_m(\vec{r}) * O(\vec{r}) = O(\vec{r}) * P_m(\vec{r}) \quad (21)$$

Where P_m is the PSF for the m -th realization of the dynamic scatterer. The crucial insight presented in our work is that, due to the commutativity of the convolution operation, the equation for the dynamic medium case has exactly the same mathematical form as the random illumination case of

conventional reflection matrix acquisition, but with the roles of the object and PSF interchanged.

Due to this mathematical equivalence of the imaging equation between the dynamic scattering and random illumination scenarios (Supplementary Eqs. 17 and 21), the covariance matrix of the acquired dataset 'measurement matrix' A has the same TDT structure as that of the covariance matrix of CTR-CLASS, and thus the I-CLASS algorithm can be used to reconstruct the object by naively applying it on the covariance matrix.

We note that to achieve a fixed object illumination throughout the different measurements in the coherent imaging case, we utilized an illumination spot size at the scattering layer plane that is smaller than the correlation length of the diffuser (see Supplementary Section 5 on "Effect of illumination spot size").

I-CLASS algorithm

The I-CLASS algorithm [28] extends the CLASS algorithm [5] estimation of phase distortions to allow both phase and amplitude distortions in conventional reflection matrix measurements. Specifically, in I-CLASS the phase corrections are found in the exact same manner as the CLASS algorithm. Following the standard CLASS phase-correction, I-CLASS estimates the amplitude distortions in the Fourier domain (the Modulation Transfer Function, MTF) from the diagonal of the covariance matrix in the Fourier domain. This amplitude information is used to correct for the amplitude aberrations by Fourier-reweighting.

In the conventional case of a static medium using multiple illuminations [28], the I-CLASS algorithm produces an estimate of the Fourier phase and amplitude of the scattering PSF. However, in the case of a dynamic scattering medium considered in this work, applying the I-CLASS algorithm (without any change on the data or the algorithm) produces the Fourier phase and amplitude of the target object. All that is left is to Fourier transform (or digitally propagate) the algorithm output to reconstruct the image of the object at the target plane.

We structured this section to gradually introduce the concepts, starting from the basic principles of reflection matrices and progressing to our specific implementation for dynamic scattering. We believe these additions significantly improve the paper’s accessibility and provide readers with all the necessary background to understand and implement our method. We would be happy to add any additional information or clarification.

2.

Reviewer Comment

Another point that is likely to make the life on a non-specialist unnecessarily hard is that eq. 1 is only valid within the isoplanatic patch. For objects larger than the isoplanatic patch nothing of what is presented here will work. This is only briefly addressed at the end of the Discussion section (largely swiping it under the carpet) but if not tackled at the beginning is likely to confuse people.

Response

We appreciate the reviewer’s concern about the isoplanatic patch limitation. To address this important point and ensure clarity for non-specialist readers, we have added an explicit note in the Principle section immediately following Equation 1, reiterating this constraint:

”It is important to note that this convolution model is strictly valid only for objects within an isoplanatic patch. All experiments in this work were designed within this constraint. Potential extensions to larger fields of view or thick complex media are discussed in the Discussion section.”

We believe that this addition ensures that readers are aware of this limitation from the outset, helping non-specialists better understand the current limited applicability of our approach (as most reflection-matrix works to date), while referring them to the potential extensions to anisoplanatic scenarios discussed in the Discussion section.

A few more minor points:

3.

Reviewer Comment

In the introduction the authors claim that iterative phase retrieval lacks guaranteed convergence, which is technically not true. Convergence might take a VERY long time, but it will eventually happen (see <https://doi.org/10.1364/A0.21.002758>).

Response

We thank the reviewer for this correction. We have revised the text to more accurately reflect the nature of iterative phase retrieval, acknowledging that convergence is possible, albeit potentially time-consuming. The revised sentence now reads:

”However, despite this advantage, these techniques are hindered by their reliance on iterative phase retrieval [6], which can require a very large number of iterations to converge, as well as specific support priors and potentially a large number of initial guesses.”

4.

Reviewer Comment

Just below the above statement there are two sentences beginning with ”while” which looks like they are the leftover of some copy-paste during editing.

Response

We thank the reviewer for their careful reading. We have removed the redundant sentence and simplified the text to improve clarity and flow. The revised text now reads:

”While deterministic bispectrum reconstruction can address the convergence challenge of phase retrieval, it still requires averaging a large number of speckle grains, limiting the reconstruction to relatively simple objects.”

5.

Reviewer Comment

The results shown in Fig. 2 and 3, albeit impressive, require the target to be illuminated from behind, which makes this whole approach invasive. I know the author never explicitly claim non-invasiveness, but they also never make this point clear.

Response

We thank the reviewer for highlighting this point. We agree that the initial results presented in Figures 2 and 3 require illumination from behind the sample, which makes the presented experiments invasive. However, it is worth noting that the addition of a scattering medium between the light source and the target in these experiments would not change the imaging performance as long as sufficiently high intensity is passed through such a scattering medium, as the incoherent imaging configuration only requires homogeneous illumination of the target.

To clarify this explicitly in the revised manuscript, we added the following note to the results section:

”Note that although these initial experiments utilize a transmission geometry with illumination from behind the target, the addition of a scattering medium between the light source and the target in these experiments would not change the imaging performance as long as a sufficiently high intensity is passed through the scattering medium, as the incoherent imaging configuration only requires homogeneous illumination of the target. Additionally, our subsequent experimental demonstrations in epi-illumination and detection in fluorescence microscopy (Fig.4) and coherent holographic imaging Fig. 5) demonstrate applicability to noninvasive imaging across diverse optical configurations and imaging modalities.”

6.

Reviewer Comment

I might have missed it, but I don't think I have seen any discussion about the time needed for the I-CLASS algorithm to converge and give the claimed results.

Response

We thank the reviewer for raising this point, which was indeed not adequately discussed in the original manuscript. Following the reviewer's comment, we have added to the revised manuscript the information on the specific runtime for each iteration for the presented results on our system. This information was added to the **Materials and methods**, and it reads:

”The run time of the I-CLASS algorithm on a commercially available GPU (Nvidia RTX4090, 24 GB) was approximately $\sim 8ms$ per iteration for 150 camera frames at a resolution of 300×300 pixels and around $\sim 70ms$ per iteration for 150 camera frames at a resolution of 850×850 pixels. With our standard protocol of 1000 iterations, this yields total processing times of approximately 8 seconds and 70 seconds, respectively.”

7.

Reviewer Comment

For the results shown in Fig. 5 one needs to know the distance between the object and the scattering medium, as only the field at the scattering layer can be reconstructed, and one needs to know how far to propagate it back.

Response

We thank the reviewer for this raising this point, which was perhaps not properly explained in the original manuscript. Since the I-CLASS algorithm reconstructs the complex field at the scattering layer plane, we can digitally back-propagate it to any desired plane.

In practice, once the reconstructed field is obtained in the scattering layer plane, we digitally back-propagate it to several distances to find the plane where the object is at best focus. This "digital autofocus" capability is a fundamental strength of coherent computational imaging, analogous to the focusing capability in digital holographic microscopy but applied to imaging through scattering media. Importantly, it eliminates the need for precise a priori knowledge of the object-diffuser distance during data acquisition or during I-CLASS iterations, and can allow imaging multiple planes, as was demonstrated in previous works (e.g. Haim et al.[9]).

To address this point in the revised manuscript, we have added a new supplementary section titled "Digital Autofocus by Fresnel Propagation" that explains this capability and includes a new figure presenting reconstructions in out-of-focus planes.

4 Digital autofocus by Fresnel propagation

In holographic imaging through scattering media, a key advantage is the ability to numerically propagate the reconstructed complex field to any desired plane once the field is retrieved at a single reference plane. This capability, often termed digital autofocus, eliminates the necessity of knowing the exact object distance during data acquisition or during the I-CLASS iterations.

In our coherent imaging experiments (Fig. 5), the I-CLASS algorithm reconstructs the complex-valued object field at the scattering layer plane ($z = 0$). This field can be digitally propagated to any desired plane using the Fresnel propagation operator (or angular spectrum propagator) described in the Methods section of the main text ("Fresnel propagation via Fourier-domain transfer function").

By computationally varying the propagation distance and observing the resulting reconstructed intensity distributions, we identify the optimal object plane where the finest features of the target come into focus. This process is analogous to the physical process of adjusting the focus in a conventional microscope, but performed entirely in post-processing.

Figure S6 demonstrates this capability by showing the reconstructed field intensity at three distinct propagation distances: before the object plane ($z = 5.3$ cm), at the object plane where optimal focus is achieved ($z = 7.15$ cm), and after the object plane ($z = 9$ cm). The sharp focus observed at $z = 7.15$ cm confirms that this is indeed the correct object plane, having the highest resolution of the fine features of the USAF target.

Figure S6: **Digital autofocus capability of the coherent imaging reconstruction.** The I-CLASS algorithm provides the complex-valued object field at the scattering layer plane. This field can be back-propagated to any desired distance from the scattering layer, allowing to find the target object plane by 'digital focusing' in post-processing, without prior knowledge of the target position. Presented images show reconstructed field intensity at three propagation distances: **a** $z = 5.3$ cm (before the object plane), **b** $z = 7.15$ cm (at the object plane, where optimal focus is achieved), and **c** $z = 9.0$ cm (after the object plane). Scale bars: 1 mm.

8.

Reviewer Comment

I am not sure I understand how focusing the illumination on the scattering medium can reduce the fluctuations of E_m^{ill} . Wouldn't it actually maximise them?

Response

We thank the reviewer for raising this question, which was not discussed in sufficient detail in the original manuscript.

To address this question comprehensively, we have added a new section, section 5 ("Effect of illumination spot size on coherent imaging through dynamic scattering") to the supplementary

material of the revised manuscript. The new supplementary section contains a thorough explanation and a set of numerical simulations, which demonstrate and study the limitations of the effect of focusing the illumination on the scattering layer surface. For convenience, we reproduce the full section below. In a nutshell, when the illumination beam is focused to a spot size that is sufficiently smaller than the coherence area of the scattering layer, it effectively experiences propagation through a single coherence area with a nearly constant phase function, and thus does not experience scattering, providing a homogeneous illumination of the target.

5 Effect of illumination homogeneity and stability on coherent imaging through dynamic scattering

In our coherent imaging demonstration (Fig. 5 of the main text), maintaining a constant homogeneous illumination pattern at the object plane despite the dynamic scattering introduced by the rotating diffuser is important to ensure the 'fixed object' assumption of our model (Eqs. 6-7 of the main text). While obtaining a constant homogeneous illumination through a dynamic scatterer is rather straightforward with spatially-incoherent illumination, it is often challenging when spatially coherent illumination is considered. Nonetheless, it can be achieved for the case of a dynamic *thin* scatterer, such as a diffuser, by focusing the illumination spot size on the diffuser, such that the spot size is sufficiently smaller than the diffuser's coherence (or correlation) area. In this case, the illumination beam effectively experiences propagation through a single coherence area with a nearly constant phase function, and thus does not experience scattering, providing a homogeneous illumination of the target. Fig. S7 demonstrates and numerically studies the limitation of this approach. The top row (Fig. S7a-c) shows the phase pattern of a thin scattering layer with the illumination spot superimposed for three cases: where the illumination spot diameter is 0.1, 0.3, and 0.5 times the diffuser correlation length ($d_{\text{correlation}}$), respectively. The second row (Fig. S7d-f) shows the resulting illumination intensity pattern at the object plane after the light has propagated through the diffuser, simulated using angular spectrum propagation. The third row (Fig. S7g-i) shows the effect of these illumination patterns

on the effective object (the product of the object and the illumination), while the fourth row (Fig. S7j-l) shows the reconstruction results.

Figure S7: **Effect of illumination pattern stability on coherent image reconstruction.** Numerical simulations of coherent matrix-based imaging through a thin dynamic scattering layer, studying the effect of illumination stability achieved by focusing the illumination spot size at the scattering layer surface (d_{spot}) to a size that is smaller than the scattering layer correlation length (d_{corr}). Focusing the illumination beam to a size considerably smaller than the coherence area of the scattering layer (a) results in a homogeneous illumination pattern at the target object plane (d) that remains constant while the scattering layer dynamically changes. Each column represents a different ratio of the illumination spot size to the scattering layer correlation length: $\frac{d_{\text{spot}}}{d_{\text{corr}}} = 0.1$ (left), $\frac{d_{\text{spot}}}{d_{\text{corr}}} = 0.3$ (middle), and $\frac{d_{\text{spot}}}{d_{\text{corr}}} = 0.5$ (right). **a-c** The thin scattering layer ‘phase screen’ pattern (color), with the illumination spot size highlighted. **d-f** Resulting illumination amplitude patterns at the object plane after numerical propagation through the scattering layer and free space. The grayscale colorbar indicates normalized intensity. **g-i** Effective reflected field amplitude at the object plane (i.e., the target object reflectivity profile multiplied by the illumination pattern). **j-l** I-CLASS reconstructed images from 150 dynamic scattering realizations. The small illumination spot size relative to the scattering layer coherence area (left column) maintains a relatively uniform illumination at the object plane, enabling high-quality reconstruction, while larger spot sizes (middle and right columns) result in varying speckle illumination, degrading direct reconstruction quality.

When the illumination spot is much smaller than the scattering layer coherence area (Fig. S7a), it effectively experiences a nearly constant phase, providing a relatively uniform illumination at the object plane. This consistency across different realizations of the scattering layer is required to maintain the assumption of our imaging model. Indeed, the I-CLASS reconstruction under these conditions is of rather high quality. Note that there exists a global phase shift of the illumination, but this global phase shift can be mathematically contained in the detection PSF, keeping the effective field reflected from the object fixed.

In contrast, when the illumination spot size is comparable to or larger than the diffuser correlation length (Fig. S7c), the beam simultaneously samples multiple uncorrelated regions of the diffuser. This produces complex speckle patterns at the object plane that vary significantly between diffuser positions, creating a different illumination pattern for each diffuser realization. This variation violates our assumption of constant illumination in Eqs. 6-7 of the main text. As a result, the reconstruction quality gradually degrades as the spot size increases relative to the scattering layer coherence area.

In our experimental implementation described in the main text, we carefully focused the beam to ensure a spot size smaller than the diffuser's correlation length, thereby maintaining sufficiently constant spatial illumination patterns at the object plane across different realizations, as required.

9.

Reviewer Comment

I am slightly confused by the setup diagram in Fig. 5: The role of the polarizing beam splitter near the scattering medium is clear, as it allows to reject part of the unscattered light, thus maximizing the amount of useful signal, but the polarizing beam splitter closer to the camera seems to ensure that the signal and the reference have opposite polarizations, and thus can never interfere. Am I missing something, or is the diagram wrong?

Response

We thank the reviewer for spotting this confusion. There was indeed a mistake in our original sketch. The beam splitter should have been a non-polarizing beam splitter (BS), and not a polarizing beam splitter (PBS). We have corrected this in the revised manuscript. The corrected Figure and Figure caption now reads:

Figure 5: **Experimental coherent reflection-imaging through dynamic scattering.** **a** Experimental setup: A reflective target is illuminated through a dynamically rotating scattering diffuser. $M = 180$ reflected light fields are holographically recorded in an off-axis holography configuration using a reference arm. **b** Example of the recorded distorted fields after computational propagation to the object plane. **c** One example of the recorded field intensity after computational propagation to the object plane. **d** Reconstructed object intensity at the object plane after applying the I-CLASS algorithm to compensate for scattering, followed by numerical propagation to the target plane (see Supplementary Fig. S6). **e** Complex-valued field amplitude PSFs (APSFs) estimated from each captured field. **f** Reference intensity image of the object without the diffuser present. Scale bars, $1mm$

Reviewer Comment

The comparison between I-CLASS and the phase retrieval algorithms in the supplementary information seems weird. The fact that the phase retrieval is shown to never be able to reconstruct any image more complex than a few dots looks too bad to be true, and is at odds with my personal experience.

Response

We thank the reviewer for raising this concern about the comparison between I-CLASS and the reconstruction by speckle-correlation based phase-retrieval following Katz et al.[15]. The main reason for the low quality "phase-retrieval" results is that the comparison shown in the supplementary figure is between I-CLASS and speckle correlation imaging that uses phase retrieval on an **estimated** autocorrelation, and not the standard phase retrieval on the true object power spectrum: conventional phase retrieval operates directly on the exact power spectrum of an object, whereas speckle correlation imaging first requires estimating the object's autocorrelation (or equivalently its power spectrum) from multiple scattered light patterns, generally following Antoine Labyrie's Stellar Speckle Interferometry (1970)[18]. The quality of this estimation is limited by statistical estimation noise due to the finite number of captured speckle grains (and in experiments also by photon shot noise and other noise sources) and thus affects the phase retrieval results.

The fact that the autocorrelation (or object power spectrum) is estimated from a set of distorted speckle patterns requires very specific and tailored signal conditioning, including envelope estimation and correction, windowing and background subtraction, as well as fine-tuning of the phase-retrieval reconstruction parameters including support constraints, number of iterations, choice of beta parameter, number of independent runs with random initial guesses, to name a few. Without a careful optimization of this rather large set of parameters, the reconstruction fidelity is unfortunately as poor as shown in our original figure. This is in contrast to the I-CLASS reconstruction algorithm, which has no free parameters except for the number of iterations, which does not have a very strong effect on the reconstruction fidelity.

We acknowledge that in our original manuscript, we have not provided a sufficiently clear or detailed explanation for this important point, nor have we made a large effort to carefully tweak and optimize the speckle-correlation estimation and phase retrieval reconstruction. In addition, we have added to the supplementary figure the result of phase retrieval applied directly to the target object reference image power spectrum for comparison.

While our enhanced speckle correlation implementation shows significantly improved results com-

pared to our original submission, I-CLASS still demonstrates superior reconstruction quality, even when compared to the best reconstruction obtained from 500 independent runs of the phase-retrieval algorithm on the estimated power spectrum. Most importantly, the I-CLASS matrix-based algorithmic reconstruction is extremely stable, providing solid results in every run, does not require any free parameters (e.g., choice of betas, iterative approach), signal conditioning (envelope correction, windowing, etc.), multiple runs with different initial guesses, or other tweaks that the phase-retrieval requires for achieving a reasonable reconstruction.

The updated supplementary section and supplementary figure that reflect these improvements and provide a more comprehensive comparison appear below:

3 Comparison with speckle-correlation phase-retrieval based reconstruction

In this section, we present a numerical comparison between our proposed matrix-based approach for the incoherent imaging case and the established speckle correlation phase-retrieval based imaging [4, 15]. Inspired by Labeyrie’s 1970 stellar speckle interferometry [18], speckle correlation imaging [4, 15]. is composed of two steps: the first is an estimation of the object autocorrelation (that is, its power spectrum) from the measured speckle patterns autocorrelation (or power spectrum); and the second is a subsequent reconstruction of the object from this estimated power-spectrum using a phase-retrieval algorithm [7]. Both methods utilize similar experimental setups and image acquisition schemes, allowing for a direct performance evaluation. To perform this comparison, we focused on several numerically simulated isoplanatic imaging scenarios. Our results demonstrate the superior performance of the matrix-based I-CLASS approach when reconstructing complex non-sparse natural target objects. Importantly, in contrast to the I-CLASS reconstruction algorithm that has no free parameters except for the number of iterations, the estimation of the object’s autocorrelation from measured speckle patterns and the phase retrieval algorithm require very careful optimization and tailoring of signal conditioning, including envelope estimation and correction, windowing and background subtraction, as well as fine-tuning of the phase-

retrieval reconstruction algorithm parameters, including specific support constraints, number of iterations, choice of beta parameter, and number of independent runs with random initial guesses. The reconstruction fidelity obtained by phase retrieval of speckle correlations is extremely sensitive to the exact choice of signal conditioning and parameters, and without a precise and careful optimization of this rather large set of parameters, the reconstruction fidelity may be very poor for complex non-sparse objects.

Figure S5 presents the results of this numerical study. Each of the two rows of panels in Fig. S5 displays the results for a different target object. For each target object, we display the widefield reference image of the object taken without the scattering medium present (Fig. S5a,f), a sample simulated captured frame distorted by scattering (Fig. S5b,g), the speckle correlation phase-retrieval reconstruction (Fig. S5c,h), the matrix-based I-CLASS reconstruction (Fig. S5d,i), and the result of the same phase retrieval algorithm applied directly to the reference image power spectrum (Fig. S5e,j).

In all of the cases tested, our matrix-based approach demonstrated superior reconstruction fidelity than speckle correlations phase-retrieval based reconstructions. Most importantly, the I-CLASS matrix-based algorithmic reconstruction is extremely stable, providing solid results in every run, does not require any free parameters, signal conditioning, or other tweaks that speckle-correlation phase-retrieval requires for achieving a reasonable reconstruction. Strikingly, the matrix-based reconstruction demonstrates superior reconstruction quality even when compared to the best phase-retrieval speckle-correlation reconstruction obtained from 500 independent runs of the phase-retrieval algorithm on the optimized estimated power spectrum.

For producing Supplementary Fig. S5, we have applied the implementation of I-CLASS algorithm given in Weinberg et al.[28] directly on a set of 100 simulated captured frames. These frames were generated by convolving the original target image with 100 different random simulated speckle PSFs in the Fourier domain. For the speckle-correlation reconstruction, we employed the reconstruction approach of Bertolotti et al. and Katz et

al. [4, 15] using the same set of 100 simulated distorted images, following a rigorous optimization of the choice of signal conditioning and algorithmic parameters, which are as follows:

1. Calculate the autocorrelation of each scattered light intensity frame.
2. Average all autocorrelations. The resulting autocorrelation suffers from an envelope given by the average envelope of the scattered light patterns.
3. Correct the autocorrelation envelope by dividing the result of step (2) by the autocorrelation of the mean of the images. This provides an estimate of the envelope of the PSF and normalizes this envelope from the raw average autocorrelation of all frames.
4. Subtract the minimum value of the resulting autocorrelation to account for the near constant background of 2:1 speckle intensity autocorrelation [15].
5. Apply appropriate windowing for optimal phase-retrieval results: a square-root of a Tukey window for the USAF target and a Tukey window taken to the power of $1/4$ for the camera-man target.

For phase retrieval, following Katz et al. [15], we ran 500 independent runs of the following phase-retrieval algorithms, where in each run a different random initial guess for the object was used, and the lowest error reconstruction is presented in Fig. S5,:

1. 300 iterations of HIO algorithm with a decreasing value of beta parameter, from 2.0 to 1.2, in steps of 0.2. A total of $5 \times 300 = 1,500$ iterations.
2. 300 additional iterations of error reduction algorithm.
3. In each iteration we applied the following object-dependent constraints:
 - For the cameraman: real, positive, and non-negative constraints
 - For the USAF target: same constraints plus an additional finite square support constraint.

Figure S5: **Numerical comparison of imaging through dynamic scattering media with our matricial approach vs. speckle-correlation phase-retrieval based imaging.** A comparison between the matrix-based I-CLASS reconstruction and speckle correlation phase-retrieval based imaging [4, 15]. **a,f** Simulated target objects. **b,g** Sample simulated captured single camera frames under dynamic scattering. **c,h** Best results of speckle correlation imaging reconstructions from 500 runs of HIO phase-retrieval algorithm, followed by error-reduction algorithm, after careful optimization of object autocorrelation estimation (see details in text). **d,i** I-CLASS reconstructions showing superior fidelity from every single run with no free parameters or signal conditioning. **e,j** Phase retrieval applied directly to the reference images power spectrum, for comparison. Scale bar: 10 px.

In this discussion, we have added the following references to the revised manuscript:

- [7] - Fienup, Applied Optics, 1982

References

- [1] Amaury Badon, Victor Barolle, Kristina Irsch, A Claude Boccara, Mathias Fink, and Alexandre Aubry. Distortion matrix concept for deep optical imaging in scattering media. *Science Advances*, 6(30):eaay7170, 2020.
- [2] Amaury Badon, Dayan Li, Geoffroy Lerosey, A Claude Boccara, Mathias Fink, and Alexandre Aubry. Smart optical coherence tomography for ultra-deep imaging through highly scattering media. *Science advances*, 2(11):e1600370, 2016.
- [3] Jacopo Bertolotti and Ori Katz. Imaging in complex media. *Nature Physics*, 18(9):1008–1017, 2022.
- [4] Jacopo Bertolotti, Elbert G Van Putten, Christian Blum, Ad Lagendijk, Willem L Vos, and Allard P Mosk. Non-invasive imaging through opaque scattering layers. *Nature*, 491(7423):232–234, 2012.

- [5] Wonjun Choi, Munkyu Kang, Jin Hee Hong, Ori Katz, Byunghak Lee, Guang Hoon Kim, Youngwoon Choi, and Wonshik Choi. Flexible-type ultrathin holographic endoscope for microscopic imaging of unstained biological tissues. *Nature communications*, 13(1):4469, 2022.
- [6] James R Fienup. Reconstruction of an object from the modulus of its fourier transform. *Optics letters*, 3(1):27–29, 1978.
- [7] James R Fienup. Phase retrieval algorithms: a comparison. *Applied optics*, 21(15):2758–2769, 1982.
- [8] Jérôme Gateau, Hervé Rigneault, and Marc Guillon. Complementary speckle patterns: deterministic interchange of intrinsic vortices and maxima through scattering media. *Physical Review Letters*, 118(4):043903, 2017.
- [9] Omri Haim, Jeremy Boger-Lombard, and Ori Katz. Image-guided computational holographic wavefront shaping. *Nature Photonics*, 19(1):44–53, 2025.
- [10] Paul S Idell, J R Fienup, and Ron S Goodman. Image synthesis from nonimaged laser-speckle patterns. *Optics Letters*, 12(11):858–860, 1987.
- [11] Yonghyeon Jo, Ye-Ryoung Lee, Jin Hee Hong, Dong-Young Kim, Junhwan Kwon, Myunghwan Choi, Moonseok Kim, and Wonshik Choi. Through-skull brain imaging in vivo at visible wavelengths via dimensionality reduction adaptive-optical microscopy. *Science advances*, 8(30):eabo4366, 2022.
- [12] Sungsam Kang, Pilsung Kang, Seungwon Jeong, Yongwoo Kwon, Taeseok D Yang, Jin Hee Hong, Moonseok Kim, Kyung-Deok Song, Jin Hyoung Park, Jun Ho Lee, et al. High-resolution adaptive optical imaging within thick scattering media using closed-loop accumulation of single scattering. *Nature communications*, 8(1):2157, 2017.
- [13] Sungsam Kang, Yongwoo Kwon, Hojun Lee, Seho Kim, Jin Hee Hong, Seokchan Yoon, and Wonshik Choi. Tracing multiple scattering trajectories for deep optical imaging in scattering media. *Nature communications*, 14(1):6871, 2023.
- [14] Sungsam Kang, Seokchan Yoon, and Wonshik Choi. Implementation of reflection matrix microscopy: an algorithm perspective. *Journal of Physics: Photonics*, 7(2):023002, feb 2025.
- [15] Ori Katz, Pierre Heidmann, Mathias Fink, and Sylvain Gigan. Non-invasive single-shot imaging through scattering layers and around corners via speckle correlations. *Nature photonics*, 8(10):784–790, 2014.

- [16] Ori Katz, Eran Small, and Yaron Silberberg. Looking around corners and through thin turbid layers in real time with scattered incoherent light. *Nature photonics*, 6(8):549–553, 2012.
- [17] Yongwoo Kwon, Jin Hee Hong, Sungsam Kang, Hojun Lee, Yonghyeon Jo, Ki Hean Kim, Seokchan Yoon, and Wonshik Choi. Computational conjugate adaptive optics microscopy for longitudinal through-skull imaging of cortical myelin. *Nature Communications*, 14(1):105, 2023.
- [18] Antoine Labeyrie. Attainment of diffraction limited resolution in large telescopes by fourier analysing speckle patterns in star images. *Astronomy and Astrophysics, Vol. 6, p. 85 (1970)*, 6:85, 1970.
- [19] Hojun Lee, Seokchan Yoon, Pascal Loohuis, Jin Hee Hong, Sungsam Kang, and Wonshik Choi. High-throughput volumetric adaptive optical imaging using compressed time-reversal matrix. *Light: Science & Applications*, 11(1):16, 2022.
- [20] Orly Liba, Matthew D Lew, Elliott D SoRelle, Rebecca Dutta, Debasish Sen, Darius M Moshfeghi, Steven Chu, and Adam de La Zerda. Speckle-modulating optical coherence tomography in living mice and humans. *Nature communications*, 8(1):15845, 2017.
- [21] Jerome Mertz, Hari Paudel, and Thomas G Bifano. Field of view advantage of conjugate adaptive optics in microscopy applications. *Applied optics*, 54(11):3498–3506, 2015.
- [22] Christopher A Metzler, Felix Heide, Prasana Rangarajan, Muralidhar Madabhushi Balaji, Aparna Viswanath, Ashok Veeraraghavan, and Richard G Baraniuk. Deep-inverse correlography: towards real-time high-resolution non-line-of-sight imaging. *Optica*, 7(1):63–71, 2020.
- [23] Ulysse Najar, Victor Barolle, Paul Balondrade, Mathias Fink, Claude Boccara, and Alexandre Aubry. Harnessing forward multiple scattering for optical imaging deep inside an opaque medium. *Nature Communications*, 15(1):7349, 2024.
- [24] ChulMin Oh, Herve Hugonnet, Moosung Lee, and YongKeun Park. Digital aberration correction for enhanced thick tissue imaging exploiting aberration matrix and tilt-tilt correlation from the optical memory effect. *Nature Communications*, 16(1):1685, 2025.
- [25] Ofer Salhov, Gil Weinberg, and Ori Katz. Depth-resolved speckle-correlations imaging through scattering layers via coherence gating. *Optics letters*, 43(22):5528–5531, 2018.
- [26] Elad Sunray, Gil Weinberg, Moriya Rosenfeld, and Ori Katz. Beyond memory-effect matrix-based imaging in scattering media by acousto-optic gating. *APL Photonics*, 9(9):096112, 09 2024.

- [27] Gil Weinberg, Munkyu Kang, Wonjun Choi, Wonshik Choi, and Ori Katz. Ptychographic lensless coherent endomicroscopy through a flexible fiber bundle. *Optics Express*, 32(12):20421–20431, 2024.
- [28] Gil Weinberg, Elad Sunray, and Ori Katz. Noninvasive megapixel fluorescence microscopy through scattering layers by a virtual incoherent reflection matrix. *Science Advances*, 10(47):eadl5218, 2024.
- [29] Seokchan Yoon, Hojun Lee, Jin Hee Hong, Yong-Sik Lim, and Wonshik Choi. Laser scanning reflection-matrix microscopy for aberration-free imaging through intact mouse skull. *Nature communications*, 11(1):5721, 2020.